# Geometric Neural Process Fields

**Wenzhe Yin[1], Zehao Xiao[1], Jiayi Shen[1]***, **Yunlu Chen[2], Cees G. M. Snoek[1],**
**Jan-Jakob Sonke[3], Efstratios Gavves[1]**
[1] *University of Amsterdam*    [2] *King Abdullah University of Science and Technology*
[3] *The Netherlands Cancer Institute*

**Reviewed on OpenReview:** *https: // openreview. net/ forum? id= yvGkEB3C26*

## Abstract

Neural Fields (NeFs) offer compact, continuous models of signals and scenes. However, one of the major issues is that a separate neural network has to be trained from scratch for each signal. To tackle this, we propose Geometric Neural Process Fields (G-*NPF*), a probabilistic framework for neural radiance fields that explicitly captures uncertainty. We formulate NeF generalization as a probabilistic problem, enabling direct inference of NeF function distributions from limited context observations. To incorporate structural inductive biases, we introduce a set of geometric bases that encode spatial structure and facilitate the inference of NeF function distributions. Building on these bases, we design a hierarchical latent variable model, allowing G-*NPF* to integrate structural information across multiple spatial levels and effectively parameterize INR functions. This hierarchical approach improves generalization to novel scenes and unseen signals. Experiments on novel-view synthesis for 3D scenes, as well as 2D image and 1D signal regression, demonstrate the effectiveness of G-*NPF* in capturing uncertainty and leveraging structural information for improved generalization.

## 1 Introduction

Neural fields (NeFs) (Sitzmann et al., 2020b; Tancik et al., 2020) have emerged as a powerful framework for learning continuous, compact representations of signals across domains, including 1D signals (Yin et al., 2022), 2D images (Sitzmann et al., 2020b), and 3D scenes (Park et al., 2019; Mescheder et al., 2019). NeFs (Vyas et al., 2024) are now used broadly in image and signal processing, including shape representation (Genova et al., 2020), novel view synthesis (Mildenhall et al., 2021), material rendering (Kuznetsov, 2021), medical imaging (Wang et al., 2022b), linear inverse problems (Sun et al., 2021), and compression (Dupont et al., 2021a). A notable advancement in 3D scene modeling is the Neural Radiance Field (NeRF) (Mildenhall et al., 2021; Barron et al., 2021), which extends NeFs to map 3D coordinates and viewing directions to volumetric density and view-dependent radiance. With differentiable volume rendering along camera rays, NeRFs enable photorealistic novel-view synthesis. Despite strong reconstruction performance, NeRFs typically require per-scene overfitting, leading to limited generalization to new scenes from few context images.

In this paper, we focus on neural field generalization (also referred to as conditional neural fields) and the rapid adaptation of NeFs to new signals. It is crucial for real-world deployment, where only sparse or constrained observations (common in robotics, AR/VR, and medical/computational imaging) are provided, requiring data-efficient adaptation with calibrated uncertainty to obviate costly per-scene optimization and to guide acquisition. Previous works on NeF generalization have addressed this challenge using gradient-based meta-learning (Tancik et al., 2021), enabling adaptation to new scenes with only a few optimization steps (Tancik et al., 2021; Papa et al., 2024). Other approaches include modulating shared MLPs through HyperNets (Chen & Wang, 2022; Mehta et al., 2021; Dupont et al., 2022a; Kim et al., 2023) or directly predicting the parameters of scene-specific MLPs (Dupont et al., 2021b; Erkoç et al., 2023). However, the deterministic nature of these methods cannot capture uncertainty in NeFs, when used with scenes with only

---

*Corresponding author.

limited observations are available. This is important as such sparse data may be interpreted in multiple valid ways.

To address uncertainty arising from having few context images, probabilistic NeFs (Gu et al., 2023; Guo et al., 2023; Kosiorek et al., 2021) have recently been investigated. For example, VNP (Guo et al., 2023) and PONP (Gu et al., 2023) infer the NeFs using Neural Processes (NPs) (Bruinsma et al., 2023; Garnelo et al., 2018b; Wang & Van Hoof, 2020), a probabilistic meta-learning method that models functional distributions conditioned on partial signal observations. These probabilistic methods, however, do not exploit potential structural information, such as the geometric characteristics of signals (e.g., object shape) or hierarchical organization in the latent space (from global to local). Incorporating such inductive biases can facilitate more effective adaptation to new signals from partial observations.

To jointly capture uncertainty and leverage inherent structural information for efficient adaptation to new signals with few observations, we propose a probabilistic neural fields generalization framework called Geometric Neural Processes Fields (G-*NPF*). Our contributions can be summarized as follows: *1) Probabilistic NeF generalization framework.* We formulate NeF generalization as a probabilistic modeling problem, allowing us to amortize a learned model over multiple signals and improve NeF learning and generalization. *2) Geometric bases.* To encode structural inductive biases, we design geometric bases that incorporate prior knowledge (e.g., Gaussian structures), enabling the aggregation of local information and the integration of geometric cues. *3) Geometric neural processes with hierarchical latent variables.* Building on these geometric bases, we develop geometric neural processes to capture uncertainty in the latent NeF function space. Specifically, we introduce hierarchical latent variables at multiple spatial scales, offering improved generalization for novel scenes and views. Experiments on 1D and 2D signals demonstrate the effectiveness of the proposed method for NeF generalization. Furthermore, we adapt our approach to the formulation of Neural Radiance Fields (NeRFs) with differentiable volume rendering on ShapeNet objects and NeRF Synthetic scenes to validate the versatility of our approach.

## 2 Background

### 2.1 Neural (Radiance) Fields

**Neural Fields (NeFs)** Sitzmann et al. (2020b) are continuous functions $f_\omega \colon x \mapsto y$, parameterized by a neural network whose parameters $\omega$ we optimize to reconstruct the continuous signal $y$ on coordinates $x$. As with regular neural networks, fitting Neural Field parameters $\omega$ relies on gradient descent minimization. Unlike regular networks, however, conventional Neural Fields are explicitly designed to overfit the signal $y$ during reconstruction deterministically, without considering generalization (Mildenhall et al., 2021; Barron et al., 2021). The reason is that Neural Fields have been primarily considered in transductive learning settings in 3D graphics, whereby the optimization objective is to optimally reconstruct the photorealism of single 3D objects at a time. In this case, there is no need for generalization across objects. A single trained Neural Field network is optimized to "fill in" the specific shape of a specific 3D object under all possible view points, given input point cloud (coordinates $x$). For each separate 3D object, we optimize a separate Neural Field afresh. Beyond 3D graphics, Neural Fields have found applicability in a broad array of 1D (Yin et al., 2022) and 2D (Chen et al., 2023b) applications, for scientific (Raissi et al., 2019) and medical data (de Vries et al., 2023), especially when considering continuous spatiotemporal settings.

**Neural Radiance Fields (NeRF)** (Mildenhall et al., 2021; Arandjelović & Zisserman, 2021) are Neural Fields specialized for 3D graphics, reconstructing the 3D shape and texture of a single objects. Specifically, each point $\mathbf{p} = (p_x, p_y, p_z)$ in the 3D space centered around the object has a color $\mathbf{c}(\mathbf{p}, \mathbf{d})$, where $\mathbf{d} = (\theta, \phi)$ is the direction of the camera looking at the point $p$. Since objects might be opaque or translucent, points also have opacity $\sigma(\mathbf{p})$. In Neural Field terms, therefore, our input comprises point coordinates and the camera direction, that is $x = (\mathbf{p}, \mathbf{d})$, and our output comprises colors and opacities, that is $y = (\mathbf{c}, \sigma)$.

Optimizing a NeRF is an inverse problem: we do not have direct access to ground-truth 3D colors and points of the object. Instead, we optimize solely based on 2D images from known camera positions $\mathbf{o}$ and viewing directions $\mathbf{d}$. Specifically, we optimize the parameters $\omega$ of the NeRF function, which encodes the 3D shape and color of the object, allowing us to render novel 2D views from arbitrary camera positions and

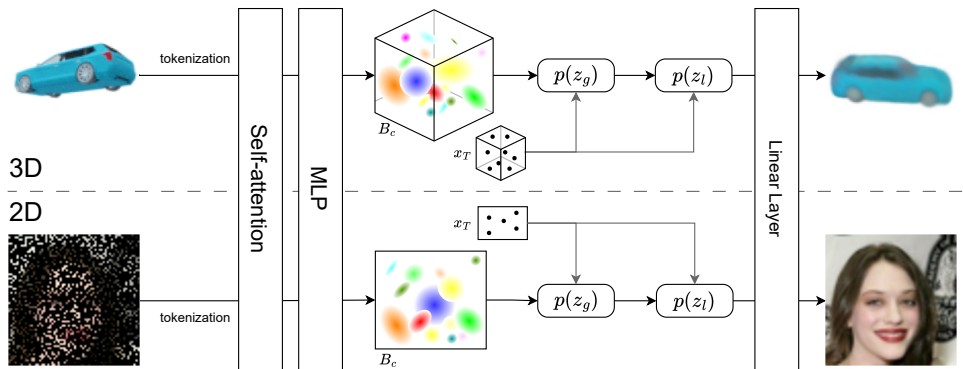

Figure 1: **Illustration of the proposed G-*NPF*.**

directions using ray tracing along $\mathbf{r} = (\mathbf{o}, \mathbf{d})$. This ray-tracing process integrates colors and opacities along the ray, accumulating contributions from points until they reach the camera. The objective is to ensure that NeRF-generated 2D views match the training images. Since these images provide an object-specific context for inferring its 3D shape and texture, we refer to them as *context data*. In contrast, all other unknown shape and texture information is *target data*. For a detailed description of the ray-tracing integration process, see Appendix B.

**Conditional Neural Fields** Papa et al. (2024) have recently gained popularity to avoid optimizing from scratch a new Neural Field for every new object. Conditional Neural Fields split parameters $\omega$ to a shared part $\omega_D$ that is common between objects in the dataset $D$, and an object-specific part $\omega_i$ that is optimized specifically for the $i$-th object. However, the optimization of $\omega_i$ is still done independently per object using stochastic gradient descent.

## 2.2  Neural Processes

Neural Processes (NPs) (Garnelo et al., 2018b; Kim et al., 2019) extend the notion of Gaussian Processes (GPs) (Rasmussen, 2003) by leveraging neural networks for flexible function approximation. Given a *context set* $\mathcal{C} = \{(x_{C,i}, y_{C,n})\}_{n=1}^{N}$ of $N$ input–output pairs, NPs infer a latent variable $z$ that captures function-level uncertainty. When presented with new inputs $x_T = \{x_{T,m}\}_{m=1}^{M}$, the goal is to predict $y_T = \{y_{T,m}\}_{m=1}^{M}$. Formally, NPs define the predictive distribution:

$$p(y_T \mid x_T, \mathcal{C}) = \int p(y_T \mid x_T, z)\, p(z \mid \mathcal{C})\, \mathrm{d}z. \tag{1}$$

Here, $p(z \mid \mathcal{C})$ is a prior over $z$ derived solely from the context set. During training, an approximate posterior $q(z \mid \mathcal{C}, \mathcal{T})$ (where $\mathcal{T}$ is the *target set* consisting of $(x_T, y_T)$ pairs) is learned via variational inference (Garnelo et al., 2018b). Through this latent-variable formulation, NPs capture both predictive uncertainty and function-level variability, enabling robust performance under partial observations.

# 3  Geometric Neural Process Fields

Despite their great reconstruction capabilities, Neural Fields are still limited by their lack of generalization. While Conditional Neural Fields offer an interesting path forward, they still suffer from the unconstrained nature of stochastic gradient descent and the over-parameterized nature of neural networks (Papa et al., 2024), thus making representation learning and generalization to few-shot settings hard, whether for 1-D (e.g., for time series), 2-D (e.g., for PINNs), or 3-D (e.g., for occupancy and radiance fields) data. We alleviate this by imposing geometric and hierarchical structure to the NeF and NeRF functions in 1-, 2-, or 3-D data, such that Neural Fields are constrained to the types of outputs that they predict. Further, we embed Conditional Neural Fields in a probabilistic learning framework using Neural Processes, so that the learned Neural Fields generalize well even with few-shot context data settings.

### 3.1 Probabilistic Neural Process Fields

Conditional Neural Fields, defined in a deterministic setting, bear direct resemblance to Neural Processes and Gaussian Processes and their context and target sets, defined in a probabilistic setting. To make the point clearer, we will use the 2D image completion task as a running example, where the goal is to reconstruct an entire image from a sparse set of observed pixels (an occluded image).

In image completion task, the $\mathcal{C} = \{(x_{C,i}, y_{C,n})\}_{n=1}^{N}$ consists of $N$ observed pixel coordinates $x_C$ and their corresponding intensity values $y_C$, while the target set $\mathcal{T} = \{x_T\}$ comprises all $M$ pixel coordinates in the image, with $y_T$ denoting the unobserved intensities to be predicted. The objective is thus to infer the full image $y_T$ conditioned on $\mathcal{C}$, effectively regressing pixel intensities across the entire spatial domain using only the sparse context observations. Although our approach is formulated as a general probabilistic framework, we present a novel 3D-specific extension for Neural Radiance Fields, detailed in Appendix G.

For probabilistic Neural Process Fields, we adopt the Neural Process decomposition from Eq. (1) for prior distribution,

$$
\begin{aligned}
p(y_T|x_T, x_C, y_C) &= \int \underbrace{p(y_T|z, x_T, x_C, y_C)}_{\text{Conditional Neural Field}} p(z|x_T, x_C, y_C)dz \\
&= \int \prod_{m=1}^{M} p(y_{T,m}|z, x_{T,m}, x_C, y_C) p(z|x_{T,m}, x_C, y_C)dz,
\end{aligned}
\tag{2}
$$

where in the last line of Eq. (2) we use the fact that the $M$ target output variables, which comprise the target object, are conditionally independent with respect to the latent variable $z$. In probabilistic Neural Process Fields, $z$ encodes object-level information, similar to the object-specific parameters $\omega_i$ in deterministic Conditional Neural Fields. However, by modeling $z$ probabilistically, our approach enables generalization across different objects, whereas standard NeFs are limited to fitting a single object at a time.

### 3.2 Adding Geometric Priors to Probabilistic Neural Process Fields

With probabilistic Neural Process Fields, we are able to generalize conditional Neural Fields to account for uncertainty and thus be more robust to few-shot learning settings. Given that (conditional) neural fields are most commonly instantiated as standard MLPs, they impose no explicit structural priors on the output and place few constraints on the types of predicted values; this flexibility can hinder generalization. While some NeRF pipelines introduce structured feature backbones (e.g., planes or grids), such designs are task- and modality-specific; there is no single unified structured representation in the NeF generalization that seamlessly covers heterogeneous signal types (e.g., 2D images and 3D radiance/shape fields).

To address this problem, we propose adding geometric priors to probabilistic Neural Process Fields. Specifically, we encode the context set $\mathcal{C}$ so that to represent it in terms of structured geometric bases $B_C = \{b\}_{r=1}^{R}$, rather than using $\mathcal{C}$ directly. Here $R$ is the number of bases. These geometric bases must create an information bottleneck through which we embed structure to the context set $\mathcal{C}$, thus $R \ll \|\mathcal{C}\| = N$. Each geometric basis $b_r = \left( \mathcal{N}(\mu_r, \Sigma_r), \omega_r \right)$ contains a Gaussian distribution $\mathcal{N}$ in the 2D spatial plane with covariance $\Sigma_r$, centered around a 2D coordinate $\mu_r$. Note that when extending to the 3D data, $\mathcal{N}$ is a 3D Gaussian. Each geometric basis also contains a representation variable $\omega_r$, learned jointly to encode the semantics around the location of $\mu_r$. The probabilistic Neural Process Field in Eq. (2) becomes

$$
p(y_T|x_T, B_C) = \int \underbrace{p(y_T|z, x_T, B_C)}_{\substack{\text{Geometric Priors on} \\ \text{Conditional Neural Fields}}} p(z|x_T, B_C)dz
\tag{3}
$$

### 3.3 Adding Hierarchical Priors to Probabilistic Neural Process Fields

The decomposition in Eq. (2) conditioning on the latent $z$ allows generalizing conditional Neural Fields with uncertainty to arbitrary training sets, especially by introducing geometric priors in Eq. (3). We note, however, that when learning probabilistic Neural Fields, our training must serve two slightly conflicting objectives. On

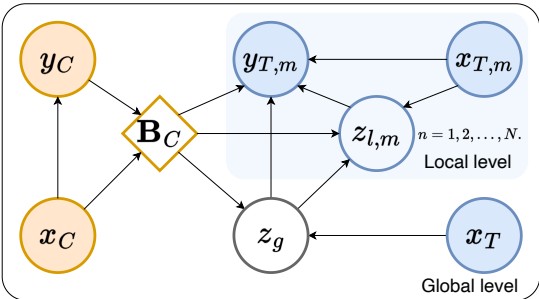

Figure 2: **Graphical model for the proposed geometric neural processes.**

one hand, the latent variable encodes the global appearance and geometry of the target object at $(x_T, y_T)$. On the other hand, the Neural Fields are inherently local, in that their inferences are coordinate-specific.

To ease the tension, we introduce hierarchical latent variables, having a single global latent variable $z_g$, and $M$ local latent variables $\{z_{l,m}\}_{m=1}^M$ for the $M$ target points $x_T$, to condition the probabilistic Neural Process Fields. A graphical model of our method is provided in Fig. 2.

$$p(y_T|x_T, B_C) = \int \int \underbrace{p(y_T|z_g, z_l, x_T, B_C)}_{\substack{\text{Hierarchical Priors on} \\ \text{Conditional Neural Fields}}} p(z_l|z_g, x_T, B_C) \ dz_l \ p(z_g|x_T, B_C) \ dz_g \tag{4}$$

$$= \int \prod_m \int p(y_{T,m}|z_g, z_{l,m}, x_{T,m}, B_C)p(z_{l,m}|z_g, x_T, B_C) \ dz_{l,m} \ p(z_g|x_T, B_C)dz_g. \tag{5}$$

In Eq. (4), we bring $p(z_g|x_T, B_C)$ out of the inside integral, which marginalizes over the local latent variables $z_l$. In Eq. (5), we further decompose by using the fact that the target variables $y_{T,m}$ and the local latent variables $z_{l,m}$ are conditionally independent.

### 3.4 Implementation

We next describe the implementation of all individual components, and refer to Appendix D for more details.

**Geometric basis functions.** We implement the geometric basis functions using a transformer encoder, $\left(\mu, \Sigma, \omega\right)_r = \texttt{Encoder}[x_C, y_C]$. In $p(z_{l,m}|z_g, x_T, B_C)$ of Eq. (5), the prior distribution of each hierarchical latent variable is conditioned on the geometric bases $B_C$ and target inputs $x_T$. Since the geometric basis functions rely on Gaussians, we use an MLP with a Gaussian radial basis function to measure their interaction:

$$\langle x_T, B_C \rangle = \texttt{MLP}\Big[\sum_{r=1}^R \exp(-\frac{1}{2}(x_T - \mu_r)^T \Sigma_r^{-1}(x_T - \mu_r)) \cdot \omega_r\Big], \tag{6}$$

**Global latent variables.** We model the global latent variable $z_g$ as a Gaussian distribution:

$$\left(\mu_g, \sigma_g\right) = \texttt{MLP}\left(\frac{1}{M}\sum_{m=1}^M \langle x_T, B_C \rangle\right), \tag{7}$$

where $p(z_g|x_T, B_C)$ is parameterized by a Gaussian whose mean $\mu_g$ and variance $\sigma_g$ are generated via an $\texttt{MLP}$. Eq (7) aggregates representations across all target points to produce a global latent variable $z_g$, thereby parameterizing the underlying object or scene. This formulation enables our model to capture object-specific uncertainty through the inferred distribution of $z_g$.

**Local latent variables.** To infer the distribution of the local latent variables $z_l$, we first compute the position-aware representation $\langle \mathbf{x}_{T,m}, B_C \rangle$ for each target point $x_{T,m}$ using Eq. (6). The local latent variable $z_{l,m}$ is then derived by combining these representations with the global latent variable $z_g$ via a transformer:

$$\left( \mu_l, \sigma_l \right) = \texttt{Transformer} \left( \texttt{MLP} \left[ \langle x_{T,m}, B_C \rangle \right]; \hat{z}_g \right),$$

where $\hat{z}_g$ is a sample from the global prior distribution $p(z_g \mid x_T, B_C)$. Mirroring the global latent variable $z_g$, we model the local prior distribution $p(z_{l,m} \mid z_g, x_{T,m}, B_C)$ as a mean-field Gaussian with parameters $\mu_l$ and $\sigma_l$. This hierarchical structure enables coordinate-specific uncertainty modeling while preserving global geometric consistency. Full architectural details are provided in Appendix D.2.

**Predictive distribution.** The hierarchical latent variables $\{z_g, z_{l,m}\}$ condition the neural network to generate predictions that integrate global and local geometric uncertainty. Specifically, the neural field is conditioned jointly on the global latent variable $z_g$, which encodes object-level structure, and the local latent variables $z_{l,m}$, which capture coordinate-specific variations. The predictive distribution $p(y_T \mid x_T, B_C)$ is obtained by propagating each target coordinate $x_{T,m}$ through the neural network, parameterized by $z_g$ and $z_{l,m}$, to model the distribution of outputs $y_{T,m}$. This process directly leverages the hierarchical prior distributions defined in Eq. (5), ensuring consistency across scales. During inference, we approximate the predictive integrals via Monte Carlo marginalization over the hierarchical latents using the learned priors conditioned on the context set. Implementation details are provided in Appendix D.3.

## 3.5 Training objective

To optimize the proposed G-*NPF*, we apply variational inference (Garnelo et al., 2018b) to derive the evidence lower bound (ELBO). Specifically, we first introduce the hierarchical variational posterior:

$$q\left( z_g, \{z_{l,m}\} \mid x_T, B_T \right) = \prod_{m=1}^{M} q\left( z_{l,m} \mid z_g, x_{T,m}, B_T \right) q\left( z_g \mid x_T, B_T \right), \tag{8}$$

where $B_T$ are target set-derived bases (available only at training). The variational posteriors are inferred from the target set $\mathcal{T}$ during training with the same encoder, which introduces more information on the object. The prior distributions are supervised by the variational posterior using Kullback–Leibler (KL) divergence, learning to model more object information with limited context data and generalize to new scenes. The details about the evidence lower bound (ELBO) and derivation are provided in the Appendix C.

Finally, the training objective includes reconstruction, hierarchical latent alignment, and geometric basis regularization:

$$\begin{aligned} \mathcal{L} = \|y_T - y_T'\|_2^2 + \alpha \Big( D_{\text{KL}} \big[ p(z_g|B_C) \,\big|\big|\, q(z_g|B_T) \big] + \sum_{m=1}^{M} D_{\text{KL}} \big[ p(z_{l,m}|z_g, B_C) \,\big|\big|\, q(z_{l,m}|z_g, B_T) \big] \Big) \\ + \beta \cdot D_{\text{KL}} \big[ B_C \,\big|\big|\, B_T \big], \end{aligned} \tag{9}$$

where $y_T'$ denotes predictions, and $\alpha$, $\beta$ balance the terms. The first term enforces local reconstruction quality, while the second ensures that the prior distributions are guided by the variational posterior using the Kullback-Leibler (KL) divergence. The third term, the KL divergence, aligns the spatial distributions of $B_C$ and $B_T$, ensuring that the context bases capture the target geometry.

## 3.6 G-*NPF* in 1D, 2D, 3D

The proposed method generalizes seamlessly to 1D, 2D, and 3D signals by leveraging Gaussian structures of corresponding dimensionality. A single global variable consistently encodes the entire signal (e.g., a 3D object or a 2D image), ensuring unified representation. For local variables, we adopt a dimension-specific formulation: in 1D and 2D signals, local variables are associated with individual spatial locations; while in 3D radiance fields, we developed a mechanism where a unique local variable is assigned to each camera ray, detailed in Appendix G. This design preserves both global coherence and local adaptability across signals.

Table 1: **Quantitative results on 2D regression.** PSNR is reported. G-*NPF* outperforms baseline methods consistently on both datasets.

|  | CelebA | Imagenette |
|---|---|---|
| Learned Init (Tancik et al., 2021) | 30.37 | 27.07 |
| TransINR (Chen & Wang, 2022) | 31.96 | 29.01 |
| **G-*NPF* (Ours)** | **33.41** | **29.82** |

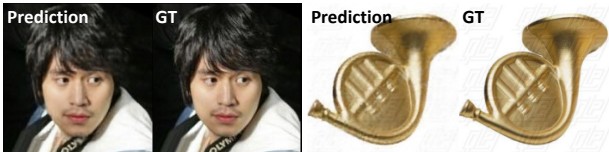

Figure 3: **Visualizations of image regression results** on CelebA (left) and Imagenette (right).

## 4 Experiments

To demonstrate the generality of G-*NPF*, we conduct extensive evaluations on, 2D image regression, 3D novel view synthesis, and 1D signal regression, comparing against recent state-of-the-art baselines.

### 4.1 G-*NPF* in 2D image regression

We start with experiments in 2D image regression, a canonical task (Tancik et al., 2021; Sitzmann et al., 2020b) to evaluate how well Neural Fields can fit and represent a 2D signal. Image regression treats an image as a continuous function $f : \mathbb{R}^2 \to [0, 1]^3$ and learns $f$ from sparse or dense pixel samples to predict color at any queried coordinate, thereby reflecting a model's capacity to fit and represent a new signal. At inference, given context pixels, we infer the mapping functions for unseen images from the test set without re-training. In this setting, the context set is an image and the task is to learn an implicit function that regresses the image pixels accurately. Following TransINR (Chen & Wang, 2022), we resize each image into $178 \times 178$, and use patch size 9 for the tokenizer. The self-attention module remains the same as our baseline, VNP (Guo et al., 2023). For the Gaussian bases, we predict 2D Gaussians. The hierarchical latent variables are inferred in image-level and pixel-level. We evaluate the method on two real-world image datasets as used in previous works (Chen & Wang, 2022; Tancik et al., 2021; Gu et al., 2023). We report peak signal-to-noise ratio (PSNR; higher is better) for comparison.

**Datasets.** We employ two real-world image datasets as used in previous works (Chen & Wang, 2022; Tancik et al., 2021; Gu et al., 2023). The CelebA dataset (Liu et al., 2015) encompasses approximately 202,000 images of celebrities, partitioned into training (162,000 images), validation (20,000 images), and test (20,000 images) sets. The Imagenette dataset (Howard, 2020), a curated subset comprising 10 classes from the 1,000 classes in ImageNet (Deng et al., 2009), consists of roughly 9,000 training images and 4,000 testing images.

**Results.** We give quantitative comparisons with respect to PSNR in Table 1. G-*NPF* outperforms baselines on both CelebA and Imagenette datasets significantly, generalizing better. Fig. 3 showcases G-*NPF*'s ability to recover high-frequency details in image regression, producing reconstructions that closely match the ground truth with high fidelity. This highlights the effectiveness of our approach. Additional qualitative results are provided in Appendix F.2.

**Image Completion.** In addition, we also conduct experiments of G-*NPF* on image completion (also called image inpainting), which is a more challenging variant of image regression. Essentially, only part of the pixels are given as context, while the INR functions are required to complete the full image. Visualizations in Fig. 4 demonstrate the generalization ability of our method to recover realistic images with fine details based on very limited context ($10\% - 20\%$ pixels).

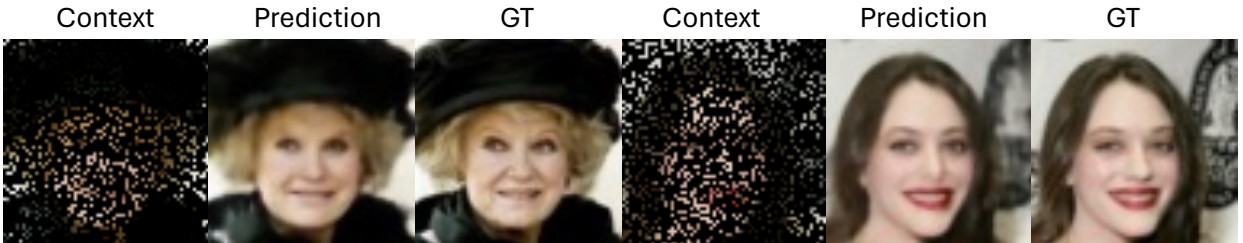

| Context | Prediction | GT | Context | Prediction | GT |

Figure 4: **Image completion visualization** on CelebA using 10% (left) and 20% (right) context.

Table 2: **Quantitative Comparison (PSNR) on Novel View Synthesis of ShapeNet Objects.** G-*NPF* outperforms baselines across categories for both 1-view and 2-view contexts. PSNR ↑ is reported.

| Method | Views | Car | Lamps | Chairs |
|---|---|---|---|---|
| Learn Init (Tancik et al., 2021) | 25 | 22.80 | 22.35 | 18.85 |
| Trans-INR (Chen & Wang, 2022) | 1 | 23.78 | 22.76 | 19.66 |
| NeRF-VAE (Kosiorek et al., 2021) | 1 | 21.79 | 21.58 | 17.15 |
| PONP (Gu et al., 2023) | 1 | 24.17 | 22.78 | 19.48 |
| VNP (Guo et al., 2023) | 1 | 24.21 | 24.10 | 19.54 |
| **G-*NPF*** (Ours) | 1 | **25.13** | **24.59** | **20.74** |
| Trans-INR (Chen & Wang, 2022) | 2 | 25.45 | 23.11 | 21.13 |
| PONP Gu et al. (2023) | 2 | 25.98 | 23.28 | 19.48 |
| **G-*NPF*** (Ours) | 2 | **26.39** | **25.32** | **22.68** |

### 4.2 G-*NPF* in 3D novel view synthesis

We continue with experiments in 3D novel view synthesis, a canonical task to evaluate 3D Neural Radiance Fields. 3D novel view synthesis (NVS) learns a continuous radiance field from a limited set of calibrated multi-view images and renders photorealistic images from previously unseen camera poses. This setting requires extrapolating colors along unseen rays while maintaining geometry and appearance consistency, thus providing a stringent benchmark of neural field generalization. We follow the implementation of Guo et al. (2023) and Chen & Wang (2022). Briefly, our input context set comprises camera rays and their corresponding image pixels from one or two views. These are split into 256 tokens, each projected into a 512D vector via a linear layer and self-attention. Two MLPs predict 256 geometric bases: one generates 3D Gaussian parameters, and the other outputs 32D latent representations. From these, we derive object- and ray-specific modulating vectors (both 512D). Our NeRF function includes four layers: two modulated and two shared layers. Further details are provided in Appendix D.1.

**ShapeNet (Chang et al., 2015).** We follow the data setup of (Tancik et al., 2021), with objects from three ShapeNet categories: chairs, cars, and lamps. For each 3D object, 25 views of size $128 \times 128$ images are generated from viewpoints randomly selected on a sphere. The objects in each category are divided into training and testing sets, with each training object consisting of 25 views with known camera poses. At test time, a random input view is sampled to evaluate the performance of the novel view synthesis. Following the setting of previous methods (Chen & Wang, 2022), we focus on the single-view (1-shot) and 2-view (2-shot) versions of the task, where one or two images with their corresponding camera rays are provided as the context.

We first compare with probabilistic Neural Field baselines, including NeRF-VAE (Kosiorek et al., 2021), PONP (Gu et al., 2023), and VNP (Guo et al., 2023). Like G-*NPF*, PONP (Gu et al., 2023) and VNP (Guo et al., 2023) also use Neural Processes, without, however, considering either geometric or hierarchical priors. Secondly, we also compare with well-established deterministic Neural Fields, including LearnInit (Tancik et al., 2021) and TransINR (Chen & Wang, 2022). We note that recent works (Liu et al., 2023; Shi et al., 2023b) have shown that training on massive 3D datasets (Deitke et al., 2023) is highly beneficial for Neural Radiance Fields. We leave massive-scale settings and comparisons to future work. Thirdly, to demonstrate

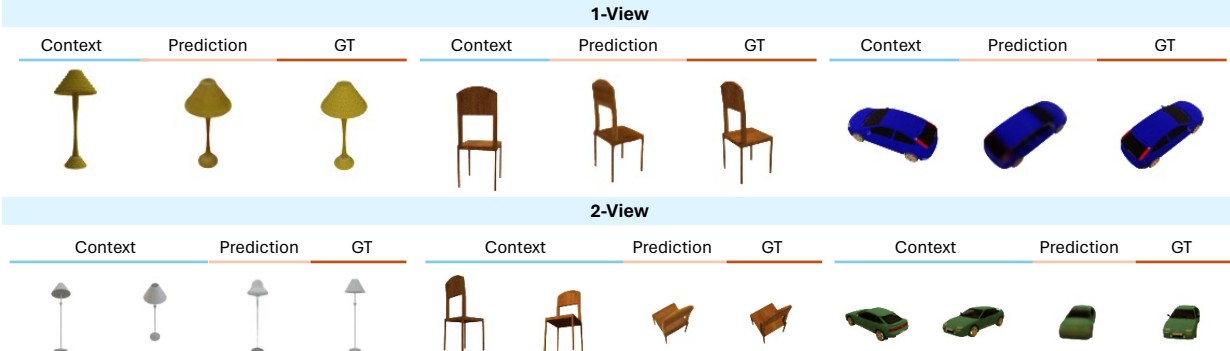

Figure 5: **Qualitative results of the proposed G-*NPF* on novel view synthesis of ShapeNet objects.** Both 1-view (top) and 2-view (bottom) context results are presented.

Table 3: **Quantitative Comparison on Novel View Synthesis of NeRF Synthetic.** G-*NPF* outperforms baselines consistently.

| Models | # Views | PSNR (↑) | SSIM (↑) | LPIPS (↓) |
|---|---|---|---|---|
| GNT (Wang et al., 2022a) | 1 | 10.25 | 0.583 | 0.496 |
| G-*NPF* | 1 | **20.07** | **0.815** | **0.208** |
| GNT (Wang et al., 2022a) | 2 | 23.47 | 0.877 | 0.151 |
| MatchNeRF (Chen et al., 2023a) | 2 | 20.57 | 0.864 | 0.200 |
| GeFu (Liu et al., 2024) | 2 | 25.30 | **0.939** | 0.082 |
| G-*NPF* | 2 | **25.66** | 0.926 | **0.081** |
| GNT (Wang et al., 2022a) | 3 | 25.80 | 0.905 | 0.104 |
| MatchNeRF (Chen et al., 2023a) | 3 | 23.20 | 0.897 | 0.164 |
| GeFu (Liu et al., 2024) | 3 | 26.99 | 0.952 | 0.070 |
| G-*NPF* | 3 | **27.85** | **0.958** | **0.068** |

the flexibility of G-*NPF* to handle complex scenes, we integrate with GNT (Wang et al., 2022a) and conduct experiments on the NeRF Synthetic dataset (Mildenhall et al., 2021).

**Quantitative results.**   We show Peak Signal-to-Noise Ratio (PSNR) results in Table 2. G-*NPF* consistently outperforms all other baselines across all categories by a significant margin. On average, G-*NPF* outperforms the probabilistic Neural Field baselines such as VNP (Guo et al., 2023), by 0.87 PSNR, indicating that adding structure in the form of geometric and hierarchical priors leads to better generalization. With two views for context, G-*NPF* improves significantly by about 1 PSNR.

**Qualitative results.**   In Fig. 5, we visualize the results on novel view synthesis of ShapeNet objects. G-*NPF* can infer object-specific radiance fields and render high-quality 2D images of the objects from novel camera views, even with only 1 or 2 views as context. More comparisons with VNP are provided in Appendix F.

**NeRF Synthetic (Mildenhall et al., 2021).**   The NeRF-Synthetic dataset contains eight object-centric scenes with complex geometry and realistic materials. For each scene, images are rendered from viewpoints randomly sampled on a hemisphere around the object. We further evaluate on the NeRF Synthetic dataset against recent state-of-the-art, including GNT (Wang et al., 2022a), MatchNeRF (Chen et al., 2023a), and GeFu (Liu et al., 2024). For a fair comparison, we use the same encoder and NeRF network architecture while integrating our probabilistic framework into GNT. Following GeFu, we assess performance in 2-view and 3-view settings.

*Quantitative results.* We present results in Table 3. We observe that G-*NPF* surpasses GeFu by approximately 1 PSNR in the 3-view setting, validating the effectiveness of our probabilistic framework and geometric bases. Moreover, we consider a challenging 1-view setting to examine the model's robustness under extremely

limited context. Both Table 3 and Fig. 9 indicate that G-*NPF* reconstructs novel views effectively with only a single view for context, in contrast to GNT that fails in this setting. We furthermore test cross-category generalization for our model and GNT without retraining, training on the `drums` category and evaluating on `lego`. As shown in Fig. 10, G-*NPF* leverages the available context information more effectively, producing higher-quality generations with better color fidelity compared to GNT. We give additional details in Appendix F.3.

## 4.3 G-*NPF* in 1D signal regression

Following the previous works' implementation (Guo et al., 2023; Kim et al., 2019), we conduct 1D signal regression experiments. 1D signal regression learns a continuous function $f : \mathbb{R} \to \mathbb{R}^K$ from samples $\{(t_i, y_i)\}$, where $t_i \in \mathbb{R}$ is a 1D coordinate (e.g., time) and $y_i \in \mathbb{R}^K$ the target (scalar when $K{=}1$). Queried at arbitrary $t$ for interpolation or extrapolation, it provides a simple task to quantify extrapolation ability and assess uncertainty calibration. We use synthetic functions drawn from Gaussian processes (GPs) with RBF and Matern kernels. This kernel selection, as advocated by Kim et al. (2022), ensures diverse function characteristics spanning smoothness, periodicity, and local variability. To evaluate performance, we adopt two key metrics: (1) context reconstruction error, quantifying the log-likelihood of observed data points (context set), and (2) target prediction error, measuring the log-likelihood of extrapolated predictions (target set). We compare with, VNP (Guo et al., 2023), CNP (Garnelo et al., 2018a), and ANP (Kim et al., 2019).

**Quantitative results.** We present a quantitative comparison with baselines in Table 4. G-*NPF* consistently outperforms the baselines across two types of synthetic data, demonstrating its effectiveness and flexibility in different signals.

Table 4: **Performance comparison on 1D signal regression.** Log-likelihoods ($\uparrow$) of the context set and target set are reported.

| Method | RBF kernel GP | | Matern kernel GP | |
|---|---|---|---|---|
| | Context | Target | Context | Target |
| CNP | $1.023 \pm 0.033$ | $0.019 \pm 0.015$ | $0.935 \pm 0.036$ | $-0.124 \pm 0.010$ |
| Stacked ANP | $1.381 \pm 0.001$ | $0.400 \pm 0.004$ | $1.381 \pm 0.001$ | $0.183 \pm 0.012$ |
| VNP | $1.377 \pm 0.004$ | $0.651 \pm 0.001$ | $1.376 \pm 0.004$ | $0.439 \pm 0.007$ |
| G-*NPF* | $1.397 \pm 0.006$ | $\mathbf{0.741 \pm 0.001}$ | $1.376 \pm 0.004$ | $\mathbf{0.545 \pm 0.009}$ |

## 4.4 Ablations

**Importance of hierarchical latent variables.** To demonstrate the effectiveness of the hierarchical nature of G-*NPF* with object-specific and ray-specific latent variables for modulation, we performed an ablation study on a subset of the Lamps dataset for fast evaluation. As shown in the last four rows in Table 5, either object-specific or ray-specific latent variable improves the performance of neural processes, indicating the effectiveness of the specific function modulation. With both $z_g$ and $z_l$, the method performs best, demonstrating the importance of the hierarchical modulation by latent variables. In addition, the hierarchical modulation also performs well without the geometric bases.

**Importance of geometric bases.** We also explore the effectiveness of the proposed geometric bases. As shown in Table 5 (rows 1 and 5), with the geometric bases, G-*NPF* performs clearly better. This indicates the importance of the 3D structure information modeled in the geometric bases,

Table 5: **Importance of geometric bases and hierarchical latent variables** on a subset of the Lamps scene synthesis (PSNR). $z_g$ and $z_l$ are global and local latent variables, respectively. ✓ and ✗ denote whether the component joins the pipeline or not.

| $B_C$ | $z_g$ | $z_l$ | PSNR ($\uparrow$) |
|---|---|---|---|
| ✗ | ✓ | ✓ | 23.06 |
| ✓ | ✗ | ✗ | 25.98 |
| ✓ | ✓ | ✗ | 26.24 |
| ✓ | ✗ | ✓ | 26.29 |
| ✓ | ✓ | ✓ | **26.48** |

which provide specific inferences of the INR function in different spatial levels. Moreover, the bases perform well without hierarchical latent variables, demonstrating their ability to construct 3D information and reduce misalignment between 2D and 3D spaces.

**Qualitative ablation of the hierarchical latent variables** In this section, we perform a qualitative ablation study on the hierarchical latent variables. As illustrated in Fig. 6, the absence of the global variable prevents the model from accurately predicting the object's outline, whereas the local variable captures fine-grained details. When both global and local variables are incorporated, G-*NPF* successfully estimates the novel view with high accuracy.

Figure 6: **Qualitative ablation of the hierarchical latent variables (global and local variables)**.

**Sensitivity to the number of geometric bases.** We further analyze the sensitivity to the number of geometric bases in the CelebA image regression and Lamps NeRF tasks. In image regression, we resize the images to $64 \times 64$ and use different patch sizes to construct 49, 169, and 484 bases. In the NeRF task, we keep the same setup as in Sec. 4.2 and construct 100, 250 bases. The results are provided in Table 6.

Table 6: **Sensitivity to the number of geometric bases** on NeRF and image regression.

|  | Image Regression | | | NeRF | |
|---|---|---|---|---|---|
| # Bases | 49 | 169 | 484 | 100 | 250 |
| PSNR (↑) | 28.59 | 33.74 | 44.24 | 24.31 | 24.59 |

With more bases, G-*NPF* achieves better performance consistently, indicating that large numbers of geometric Gaussian bases further enrich the structure information and lead to stronger predictive functions. We choose the number of bases by balancing the performance and computational costs.

## 5 Related Work

**Neural Fields (NeFs) and generalization.** Neural Fields (NeFs) map coordinates to signals, providing a compact and flexible continuous data representation (Sitzmann et al., 2020b; Tancik et al., 2020). They are widely used for 3D object and scene modeling (Chen & Zhang, 2019; Park et al., 2019; Mescheder et al., 2019; Genova et al., 2020; Niemeyer & Geiger, 2021). However, how to generalize to new scenes without retraining remains a problem. Many previous methods attempt to use meta-learning to achieve NeF generalization. Specifically, gradient-based meta-learning algorithms such as Model-Agnostic Meta Learning (MAML) (Finn et al., 2017) and Reptile (Nichol et al., 2018) have been used to adapt NeFs to unseen data samples in a few gradient steps (Lee et al., 2021; Sitzmann et al., 2020a; Tancik et al., 2021). Another line of work uses HyperNet (Ha et al., 2016) to predict modulation vectors for each data instance, scaling and shifting the

activations in all layers of the shared MLP (Mehta et al., 2021; Dupont et al., 2022a;b). Some methods use HyperNet to predict the weight matrix of NeF functions (Dupont et al., 2021b; Zhang et al., 2023). Transformers (Vaswani et al., 2017) have also been used as hypernetworks to predict column vectors in the weight matrix of MLP layers (Chen & Wang, 2022; Dupont et al., 2022b). In addition, Reizenstein et al. (2021); Wang et al. (2022a) use transformers specifically for NeRF. Such methods are deterministic and do not consider the uncertainty of a scene when only partially observed. Other approaches model NeRF from a probabilistic perspective (Kosiorek et al., 2021; Hoffman et al., 2023; Dupont et al., 2021b; Moreno et al., 2023; Erkoç et al., 2023). For instance, NeRF-VAE (Kosiorek et al., 2021) learns a distribution over radiance fields using latent scene representations based on VAE (Kingma & Welling, 2013) with amortized inference. Normalizing flow (Winkler et al., 2019) has also been used with variational inference to quantify uncertainty in NeRF representations (Shen et al., 2022; Wei et al., 2023). However, these methods do not consider potential structural information, such as the geometric characteristics of signals, which our approach explicitly models.

**Generalizable Neural Radiance Fields (NeRF).** PixelNeRF (Yu et al., 2021) is the seminal work that learns a scene prior for reconstructing NeRFs from one or a few images. Wang et al. (2022a) propose an attention-based NeRF architecture, demonstrating enhanced capabilities in capturing complex scene geometries by focusing on informative regions. Suhail et al. (2022) introduce a generalizable patch-based neural rendering approach, enabling models to adapt to new scenes without retraining. Xu et al. (2022) present *Point-NeRF*, leveraging point-based representations for efficient scene modeling and scalability. Wang et al. (2024) further enhance point-based methods by incorporating visibility and feature augmentation to improve robustness and generalization. Liu et al. (2024) propose a geometry-aware reconstruction with fusion-refined rendering for generalizable NeRFs, improving geometric consistency and visual fidelity. Recent diffusion-based 3D methods (Szymanowicz et al., 2023; Anciukevičius et al., 2024) learn data-level score priors over viewsets/IB-planes and adapt to each new scene via a diffusion process. By contrast, we adopt a meta-learning (Neural-Process) formulation that learns a task-level prior over neural fields across scenes, enabling amortized, single-pass adaptation from a sparse context to a full scene representation, i.e., few-shot generalization across scenes.

**Neural Processes.** Neural Processes (NPs) (Garnelo et al., 2018b) is a meta-learning framework that characterizes distributions over functions, enabling probabilistic inference, rapid adaptation to novel observations, and the capability to estimate uncertainties. This framework is divided into two classes of research. The first one concentrates on the marginal distribution of latent variables (Garnelo et al., 2018b), whereas the second targets the conditional distributions of functions given a set of observations (Garnelo et al., 2018a; Gordon et al., 2019). Typically, MLP is employed in Neural Processes methods. To improve this, Attentive Neural Processes (ANP) (Kim et al., 2019) integrate the attention mechanism to improve the representation of individual context points. Similarly, Transformer Neural Processes (TNP) (Nguyen & Grover, 2022) view each context point as a token and utilize transformer architecture to effectively approximate functions. Additionally, the Versatile Neural Process (VNP) (Guo et al., 2023) employs attentive neural processes for neural field generalization but does not consider the information misalignment between the 2D context set and the 3D target points (i.e., 2D pixels are radiance integrals along camera rays rather than pointwise samples of the 3D field, so occlusion, view-dependence, and depth ambiguity induce a many-to-one, projection-induced mismatch between observed contexts and 3D targets). The hierarchical structure in VNP is more sequential than global-to-local. Conversely, PONP (Gu et al., 2023) is agnostic to neural-field specifics and concentrates on the neural process perspective. Although both our approach and PONP (Gu et al., 2023) use the neural-process paradigm, our model introduces an explicit hierarchical latent structure with geometric bases. By contrast, PONP does not explicitly parameterize such structural information. In this work, we consider a hierarchical neural process to model the structural information of the scene.

# 6 Conclusion

In this paper, we addressed the challenge of Neural Field (NeF) generalization, enabling models to rapidly adapt to new signals with limited observations. To achieve this, we proposed Geometric Neural Processes (G-*NPF*), a probabilistic neural radiance field that explicitly captures uncertainty. By formulating neural field generalization in a probabilistic framework, G-*NPF* incorporates uncertainty and infers NeF function

distributions directly from sparse context images. To embed structural priors, we introduce geometric bases, which learn to provide structured spatial information. Additionally, our hierarchical neural process modeling leverages both global and local latent variables to parameterize NeFs effectively. In practice, G-*NPF* extends to 1D, 2D, and 3D signal generalization, demonstrating its versatility across different modalities.

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

## A  Appendix

## B  Neural Radiance Field Rendering

In this section, we outline the rendering function of NeRF (Mildenhall et al., 2021). A 5D neural radiance field represents a scene by specifying the volume density and the directional radiance emitted at every point in space. NeRF calculates the color of any ray traversing the scene based on principles from classical volume rendering (Kajiya & Von Herzen, 1984). The volume density $\sigma(\mathbf{x})$ quantifies the differential likelihood of a ray terminating at an infinitesimal particle located at $\mathbf{x}$. The anticipated color $C(\mathbf{r})$ of a camera ray $\mathbf{r}(t) = \mathbf{o} + t\mathbf{d}$, within the bounds $t_n$ and $t_f$, is determined as follows:

$$C(\mathbf{r}) = \int_{t_n}^{t_f} T(t)\sigma(\mathbf{r}(t))c(\mathbf{r}(t), \mathbf{d})dt, \quad \text{where} \quad T(t) = \exp\left(-\int_{t_n}^{t} \sigma(\mathbf{r}(s))ds\right). \tag{10}$$

Here, the function $T(t)$ represents the accumulated transmittance along the ray from $t_n$ to $t$, which is the probability that the ray travels from $t_n$ to $t$ without encountering any other particles. To render a view from our continuous neural radiance field, we need to compute this integral $C(\mathbf{r})$ for a camera ray traced through each pixel of the desired virtual camera.

## C  Hierarchical ELBO Derivation

Recall the hierarchical predictive distribution:

$$p(y_T \mid x_T, B_C) = \int \left[\int p(y_T \mid z_g, z_l, x_T, B_C)\, p(z_l \mid z_g, x_T, B_C)\, \mathrm{d}z_l\right] p(z_g \mid x_T, B_C)\, \mathrm{d}z_g, \tag{11}$$

and its factorized version across $M$ target points:

$$p(y_T \mid x_T, B_C) = \int p(z_g \mid x_T, B_C)\left[\prod_{m=1}^{M} \int p(y_{T,m} \mid z_g, z_{l,m}, x_{T,m}, B_C)\, p(z_{l,m} \mid z_g, x_{T,m}, B_C)\, \mathrm{d}z_{l,m}\right] \mathrm{d}z_g.$$

We introduce a *hierarchical* variational posterior:

$$q(z_g, \{z_{l,m}\} \mid x_T, B_T) = q(z_g \mid x_T, B_T) \prod_{m=1}^{M} q(z_{l,m} \mid z_g, x_{T,m}, B_T),$$

where $B_T$ are target-derived bases (available only at training). We then write the log-likelihood as

$$\log p(y_T \mid x_T, B_C) = \log \int \int p(y_T, z_g, \{z_{l,m}\} \mid x_T, B_C) \frac{q(z_g, \{z_{l,m}\} \mid x_T, B_T)}{q(z_g, \{z_{l,m}\} \mid x_T, B_T)}\, \mathrm{d}z_l\, \mathrm{d}z_g$$

$$= \log \int p(z_g \mid x_T, B_C) \frac{q(z_g \mid x_T, B_T)}{q(z_g \mid x_T, B_T)}\left[\int p(y_T, \{z_{l,m}\} \mid z_g, x_T, B_C) \frac{q(\{z_{l,m}\} \mid z_g, x_T, B_T)}{q(\{z_{l,m}\} \mid z_g, x_T, B_T)}\, \mathrm{d}z_l\right] \mathrm{d}z_g. \tag{12}$$

We first apply Jensen's inequality w.r.t. $q(z_g \mid x_T, B_T)$. This yields:

$$\log p(y_T \mid x_T, B_C) \geq \mathbb{E}_{q(z_g \mid x_T, B_T)}\left[\log \int p(y_T, \{z_{l,m}\} \mid z_g, x_T, B_C) \frac{q(\{z_{l,m}\} \mid z_g, x_T, B_T)}{q(\{z_{l,m}\} \mid z_g, x_T, B_T)}\, \mathrm{d}z_l\right]$$

$$- D_{\mathrm{KL}}(q(z_g \mid x_T, B_T) \parallel p(z_g \mid x_T, B_C)). \tag{13}$$

Inside the expectation over $z_g$, we have

$$\log \int p\big(y_T, \{z_{l,m}\} \mid z_g, x_T, B_C\big) \frac{q\big(\{z_{l,m}\} \mid z_g, x_T, B_T\big)}{q\big(\{z_{l,m}\} \mid z_g, x_T, B_T\big)} \, \mathrm{d}z_l \,.$$

We again apply Jensen's inequality w.r.t. $q(\{z_{l,m}\} \mid z_g, x_T, B_T)$, factorizing over $m$:

$$\log \int p\big(y_T, \{z_{l,m}\} \mid z_g, x_T, B_C\big) \frac{q\big(\{z_{l,m}\} \mid z_g, x_T, B_T\big)}{q\big(\{z_{l,m}\} \mid z_g, x_T, B_T\big)} \, \mathrm{d}z_l$$

$$\geq \ \mathbb{E}_{q(\{z_{l,m}\}|z_g,x_T,B_T)}\Big[\log p\big(y_T \mid z_g, \{z_{l,m}\}, x_T, B_C\big)\Big] \ - \ \sum_{m=1}^{M} D_{\mathrm{KL}}\Big(q\big(z_{l,m} \mid z_g, x_{T,m}, B_T\big) \,\|\, p\big(z_{l,m} \mid z_g, x_{T,m}, B_C\big)\Big).$$

$$(14)$$

Putting this back into Eq. (13), we arrive at the hierarchical ELBO:

$$\log p\big(y_T \mid x_T, B_C\big) \ \geq \ \mathbb{E}_{q(z_g|x_T,B_T)}\Bigg[\sum_{m=1}^{M} \mathbb{E}_{q(z_{l,m}|z_g,x_{T,m},B_T)}\big[\log p\big(y_{T,m} \mid z_g, z_{l,m}, x_{T,m}, B_C\big)\big]$$

$$- \sum_{m=1}^{M} D_{\mathrm{KL}}\Big(q\big(z_{l,m} \mid z_g, x_{T,m}, B_T\big) \,\|\, p\big(z_{l,m} \mid z_g, x_{T,m}, B_C\big)\Big)\Bigg] \qquad (15)$$

$$- \ D_{\mathrm{KL}}\Big(q\big(z_g \mid x_T, B_T\big) \,\|\, p\big(z_g \mid x_T, B_C\big)\Big).$$

The first expectation over $q(z_g \mid x_T, B_T)$ enforces global consistency and penalizes deviations from the prior $p(z_g \mid x_T, B_C)$. The second set of expectations over $q(z_{l,m} \mid z_g, x_{T,m}, B_T)$ shapes local reconstruction quality (via the log-likelihood) and penalizes deviations from the local prior $p(z_{l,m} \mid z_g, x_{T,m}, B_C)$.

Hence, the final ELBO (Eq. (15)) combines these outer and inner regularization terms with the expected log-likelihood of the target data $y_T$. This allows the model to learn coherent *global* structure as well as *local* (coordinate-specific) details in a principled way.

## D Implementation Details

### D.1 Gaussian Construction

Since 3D Gaussians represent a special case involving quaternion-based transformations, we use them here as an illustrative example for constructing geometric bases. However, the method remains consistent with the construction of 1D and 2D Gaussian geometric bases.

**Geometric Bases with 3D Gaussians.** To impose geometric structure on the context variables, we encode the context set $\{x_C, y_C\}$ into a set of $M$ *geometric bases*:

$$B_C \ = \ \Big\{ b_r \Big\}_{r=1}^{R}, \quad \text{where} \quad b_r = \ \Big(\mathcal{N}\big(\mu_r, \ \Sigma_r\big), \ \omega_r\Big). \qquad (16)$$

Each basis $b_r$ is thus defined by a 3D Gaussian $\mathcal{N}(\mu_r, \Sigma_r)$ and an associated semantic embedding $\omega_r$. The center $\mu_r \in \mathbb{R}^3$ and covariance $\Sigma_i \in \mathbb{R}^{3 \times 3}$ capture location and shape, while $\omega_r \in \mathbb{R}^{d_B}$ represents additional learned properties (e.g., color or texture). In our implementation, $d_B = 32$.

**Self-Attention Construction.** We use a self-attention module, denoted $\mathtt{Att}$, to extract these Gaussian parameters from the context data. Concretely,

$$\mu_i, \ \Sigma_i \ = \ \mathtt{Att}\big(x_C, y_C\big), \qquad \omega_i \ = \ \mathtt{Att}\big(x_C, y_C\big), \qquad (17)$$

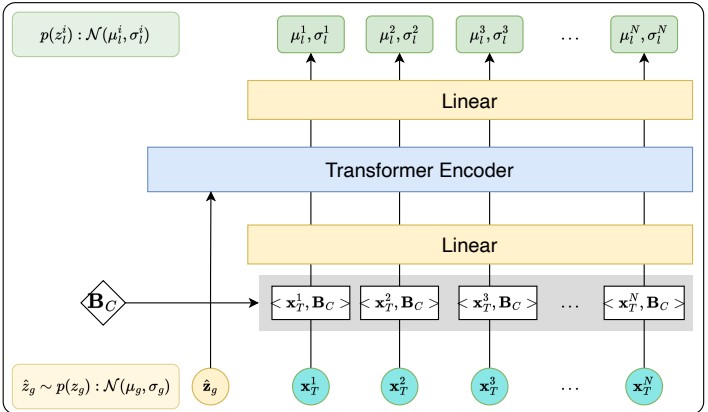

Figure 7: **Using transformer encoder to generate ray-specific latent variable $\mathbf{z}_l$.**

where each call to `Att` produces $M$ *tokens* of hidden dimension $D$. An MLP then maps each token into a 10-dimensional vector encoding: (i) the 3D center $\mu_i$, (ii) a 3D scaling vector $\mathbf{s}_i$, and (iii) a 4D quaternion $\mathbf{q}_i$ that, together, define the rotation matrix $\mathbf{R}_i$. Following Kerbl et al. (2023), we obtain the covariance $\Sigma_i$ via

$$\Sigma_i = \mathbf{R}_i \left( \mathbf{S}_i \mathbf{S}_i^\top \right) \mathbf{R}_i^\top, \tag{18}$$

where $\mathbf{S}_i = \mathrm{diag}(\mathbf{s}_i) \in \mathbb{R}^{3 \times 3}$ is the scaling matrix and $\mathbf{R}_i \in \mathbb{R}^{3 \times 3}$ is derived from $\mathbf{q}_i$. A separate MLP outputs the 32-dimensional embedding $\omega_i$. Consequently, each $b_i$ is a fully parameterized 3D Gaussian plus a semantic vector, allowing the model to represent rich geometric information inferred from the context set.

### D.2 Hierarchical Latent Variables

At the object level, the distribution of the global latent variable $z_g$ is obtained by aggregating all location representations from $(B_C, x_T)$. We assume that $p(z_g \mid B_C, x_T)$ follows a standard Gaussian distribution, and we generate its mean $\mu_g$ and variance $\sigma_g$ using MLPs. We then sample a global modulation vector, $\hat{z}_g$, from its prior distribution $p(z_g \mid x_T, B_C)$.

Similarly, as shown in Fig. 7, we aggregate information for each target coordinate $x_{T,m}$ using $B_C$, which is then processed through a Transformer along with $\hat{z}_g$ to predict the local latent variable $z_{l,m}$ for each target point. The mean $\mu_{l,m}$ and variance $\sigma_{l,m}$ of $z_{l,m}$ are obtained via MLPs.

### D.3 Modulation

We use modulation to the latent variables for modulating the MLP are represented as $[z_g; z_l]$. Our approach to the modulated MLP layer follows the style modulation techniques described in (Karras et al., 2020; Guo et al., 2023). Specifically, we consider the weights of an MLP layer (or 1x1 convolution) as $W \in \mathbb{R}^{d_{\mathrm{in}} \times d_{\mathrm{out}}}$, where $d_{\mathrm{in}}$ and $d_{\mathrm{out}}$ are the input and output dimensions respectively, and $w_{ij}$ is the element at the $i$-th row and $j$-th column of $W$.

To generate the style vector $s \in \mathbb{R}^{d_{\mathrm{in}}}$, we pass the latent variable $z$ through two MLP layers. Each element $s_i$ of the style vector $s$ is then used to modulate the corresponding parameter in $W$.

$$w'_{ij} = s_i \cdot w_{ij}, \quad j = 1, \ldots, d_{\mathrm{out}}, \tag{19}$$

where $w_{ij}$ and $w'_{ij}$ denote the original and modulated weights, respectively.

The modulated weights are normalized to preserve training stability,

$$w''_{ij} = \frac{w'_{ij}}{\sqrt{\sum_i w'^2_{ij} + \epsilon}}, \quad j = 1, \ldots, d_{\mathrm{out}}. \tag{20}$$

# E   Implementation Details

We train all our models with PyTorch. Adam optimizer is used with a learning rate of $1e-4$. For NeRF-related experiments, we follow the baselines (Chen & Wang, 2022; Guo et al., 2023) to train the model for 1000 epochs. All experiments are conducted on four NVIDIA A6000 GPUs. For the hyperparameters $\alpha$ and $\beta$, we simply set them as 0.001.

**Model Complexity**   The comparison of the number of parameters is presented in Table. 7. Our method, G-*NPF*, utilizes fewer parameters than the baseline, VNP, while achieving better performance on the ShapeNet Car dataset in terms of PSNR.

Table 7: Comparison of the number of parameters and PSNR on the ShapeNet Car dataset.

| Method | # Parameters | PSNR |
|---|---|---|
| VNP | 34.3M | 24.21 |
| G-*NPF* | **24.0M** | **25.13** |

# F   More Experimental Results

## F.1   More Metrics

We include the SSIM and LPIPS metrics for both the image regression and ShapeNet reconstruction tasks. Following the NeF generalization setting (Trans-INR (Chen & Wang, 2022); VNP (Guo et al., 2023)), which do not originally report these metrics, we computed the scores as follows: for 3D reconstruction, we used the pretrained weights provided by VNP; for 2D image regression, we re-ran the publicly available Trans-INR implementation and calculated the metrics accordingly. As shown in Table 8 and Table 9, our method consistently achieves higher SSIM and lower LPIPS scores than the baseline, indicating improved structural similarity and perceptual quality.

Table 8: Comparison of PSNR, SSIM, and LPIPS for **car**, **lamps**, and **chair** categories.

| Car | PSNR ↑ | SSIM ↑ | LPIPS ↓ |
|---|---|---|---|
| VNP (Guo et al., 2023) | 24.21 | 0.92 | 0.13 |
| Ours | **25.13** | **0.95** | **0.11** |

| Lamps | PSNR ↑ | SSIM ↑ | LPIPS ↓ |
|---|---|---|---|
| VNP (Guo et al., 2023) | 24.10 | 0.91 | 0.16 |
| Ours | **24.59** | **0.92** | **0.14** |

| Chair | PSNR ↑ | SSIM ↑ | LPIPS ↓ |
|---|---|---|---|
| VNP (Guo et al., 2023) | 19.54 | 0.83 | 0.22 |
| Ours | **20.74** | **0.87** | **0.19** |

## F.2   Image Regression

We provide more image regression results to demonstrate the effectiveness of our method as shown in Fig. 8.

## F.3   Comparison with other baselines.

**Comparison with GNT**   For fair comparison, we use GNT's image encoder and predict the geometric bases, and GNT's NeRF' network for prediction. Fig. 9 shows that our method is effective when very limited

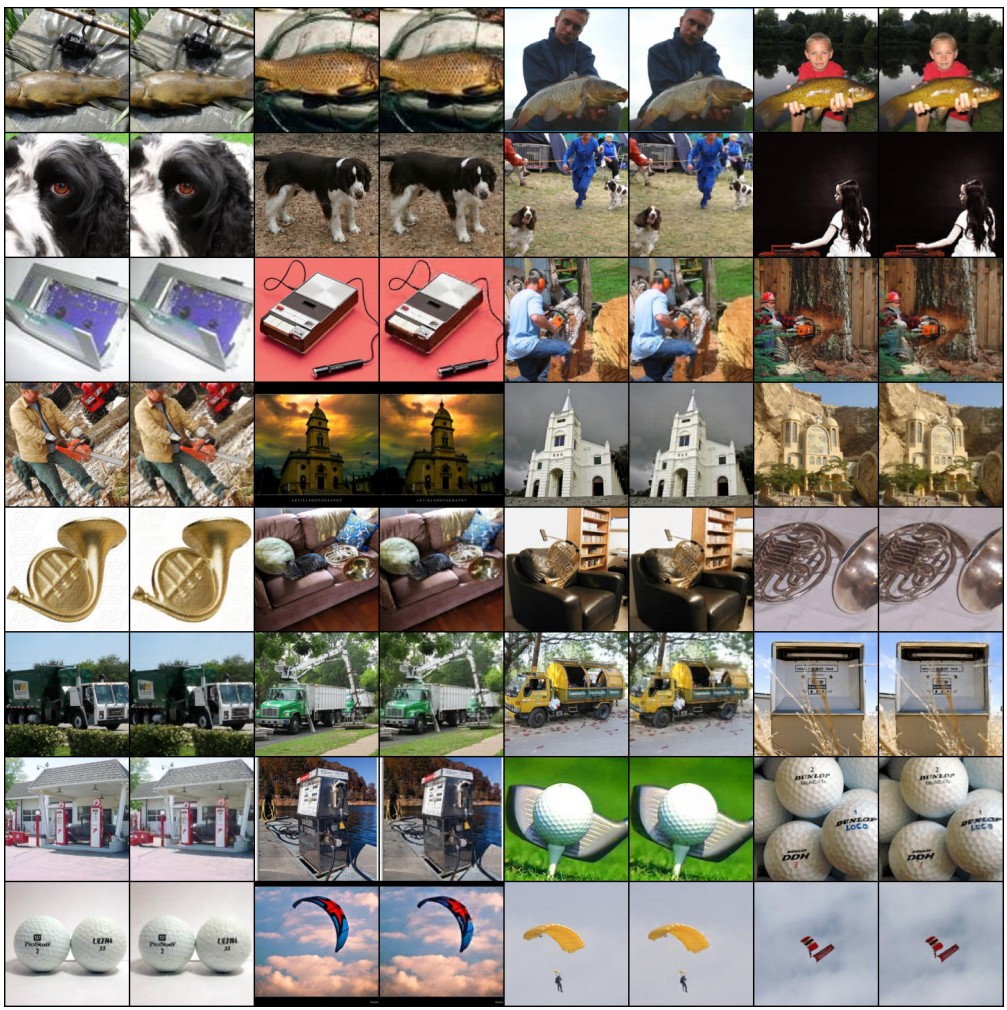

Figure 8: **More image regression results on the Imagenette dataset.** Left: ground truth; Right: prediction.

Table 9: Results on CelebA and Imagenette.

| Method | CelebA | | | Imagenette | | |
|---|---|---|---|---|---|---|
| | PSNR ↑ | SSIM ↑ | LPIPS ↓ | PSNR ↑ | SSIM ↑ | LPIPS ↓ |
| Trans-INR (Chen & Wang, 2022) | 31.96 | 0.92 | 0.07 | 29.01 | 0.86 | 0.14 |
| **Ours** | **33.41** | **0.96** | **0.04** | **29.82** | **0.93** | **0.09** |

context information is given, while GNT fails. This indicates that our method can sufficiently utilize the given information.

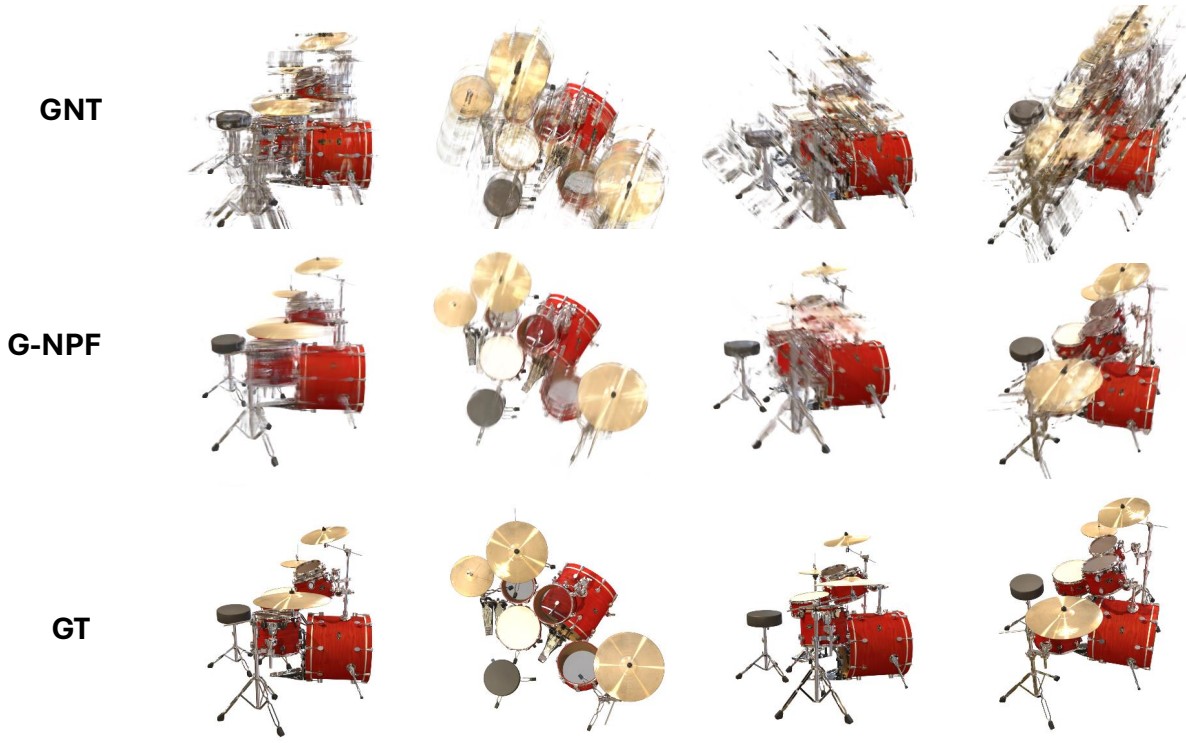

Figure 9: **Qualitative comparison with GNT on 1-view setting.**

**Cross-Category Example.** In our NeRF-synthetic experiments, "generalization" follows the standard protocol and refers to novel-view prediction within a scene rather than across scenes. Our formulation is NP-like but targets a distribution over location-specific functions (i.e., the conditional radiance field given a context set for that scene), not a distribution over functions across a large meta-dataset of scenes. For the cross-scene function distribution, we additionally perform cross-category evaluation without retraining the model. The model is trained on `drums` category and evaluated on `lego`. As shown in Figure 10, G-*NPF* leverages the available context information more effectively, producing higher-quality generations with better color fidelity compared to GNT.

**Comparison with Viewset Diffusion** We also compare against Viewset Diffusion (Szymanowicz et al., 2023). Concretely, we adopt its encoder as the backbone to estimate our geometric bases and then apply our hierarchical modulation for reconstruction. Under the ShapeNet Car setting of Viewset Diffusion (Table 10), our method achieves higher PSNR and SSIM and lower LPIPS than the baseline.

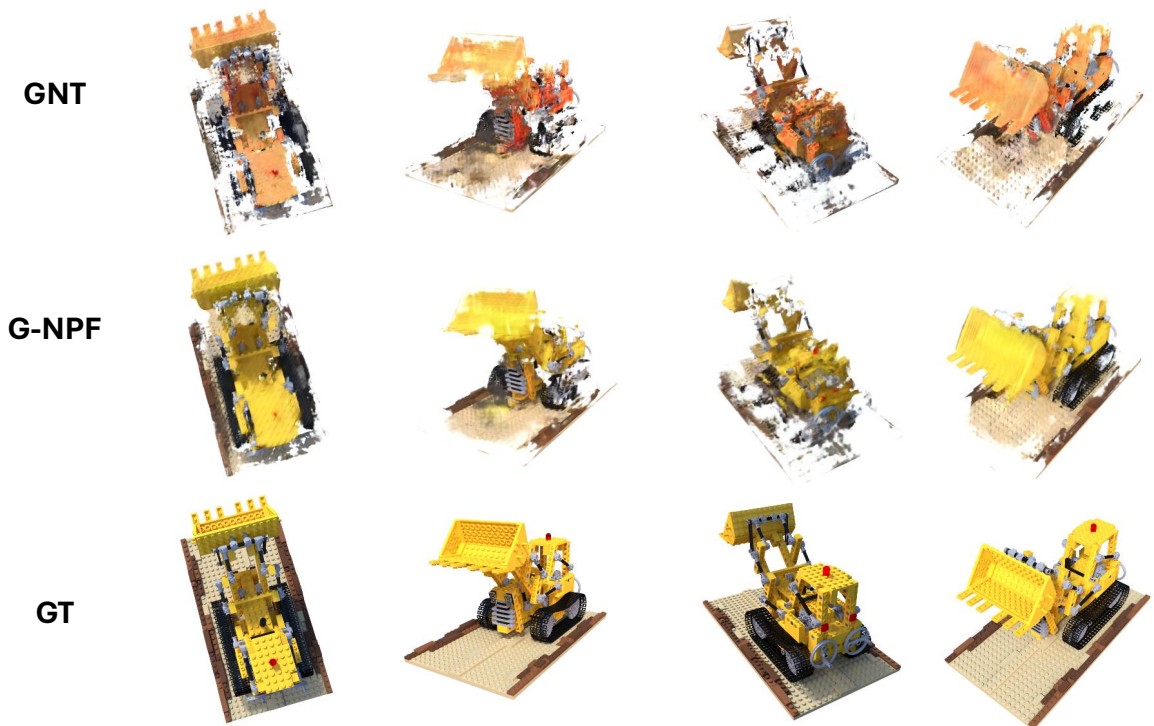

Figure 10: **Qualitative comparison of cross-category ability.** Methods are trained on the Drums category and evaluated on LEGO.

Table 10: ShapeNet Car results using the Viewset Diffusion backbone. Higher is better for PSNR/SSIM; lower is better for LPIPS.

| Method | PSNR ↑ | SSIM ↑ | LPIPS ↓ |
|---|---|---|---|
| Viewset Diffusion (Szymanowicz et al., 2023) | 28.00 | 0.871 | 0.167 |
| **Ours** | **28.68** | **0.892** | **0.153** |

### F.4 More results on ShapeNet

In this section, we demonstrate more experimental results on the novel view synthesis task on ShapeNet in Fig 11, comparison with VNP Guo et al. (2023) in Fig. 12, and image regression on the Imagenette dataset in Fig. 8. The proposed method is able to generate realistic novel view synthesis and 2D images.

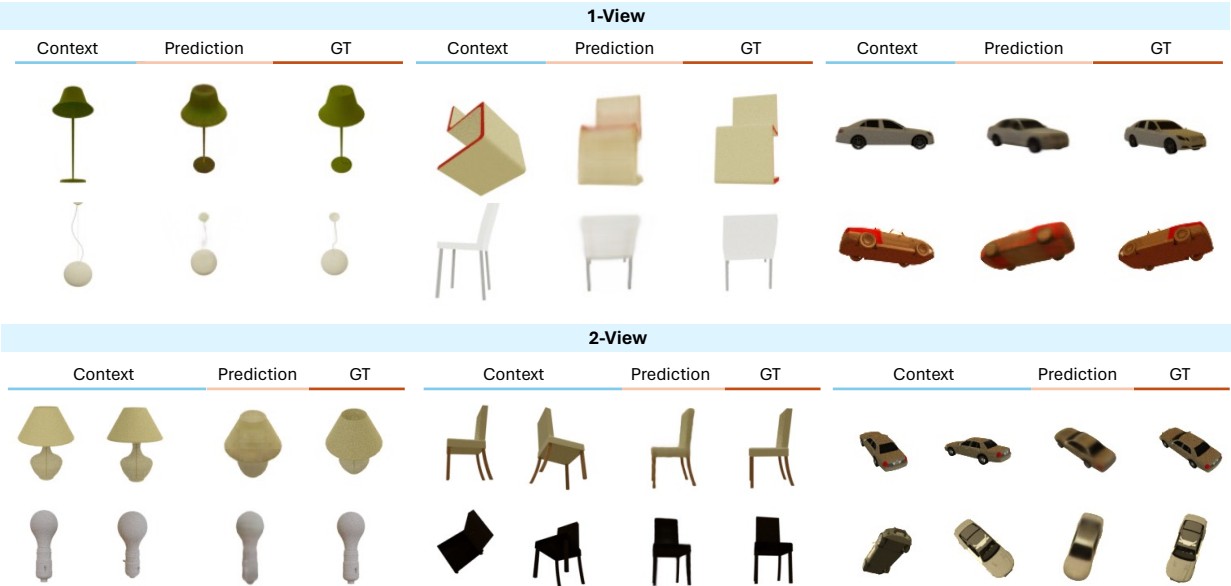

Figure 11: **More NeRF results on novel view synthesis task on ShapeNet objects.**

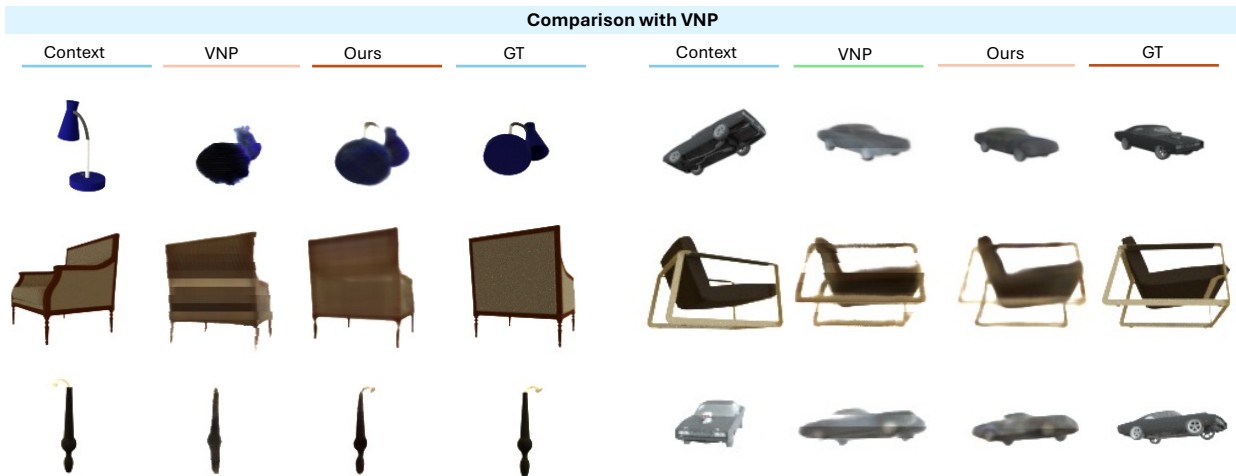

Figure 12: **Comparison between the proposed method and VNP** on novel view synthesis task for ShapeNet objects. Our method has a better rendering quality than VNP for novel views.

### F.5 Training Time Comparison

As illustrated in Fig.14, with the same training time, our method (G-*NPF*) demonstrates faster convergence and higher final PSNR compared to the baseline (VNP).

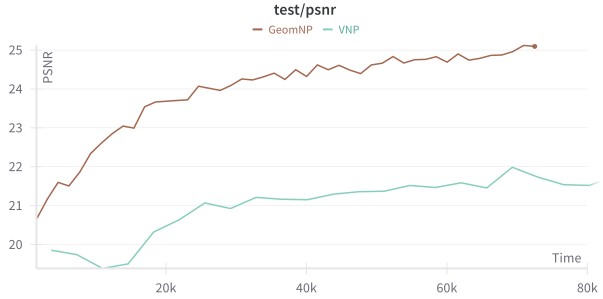

Figure 13: **Training time vs. PSNR on the ShapeNet Car dataset.** Our method (G-*NPF*) demonstrates faster convergence and higher final PSNR compared to the baseline (VNP).

## F.6 Diverse Sampling

We provide qualitative examples and quantitative results for the image completion task. The results demonstrate that our method can generate outputs that are both visually similar to the ground truth and exhibit meaningful diversity across samples.

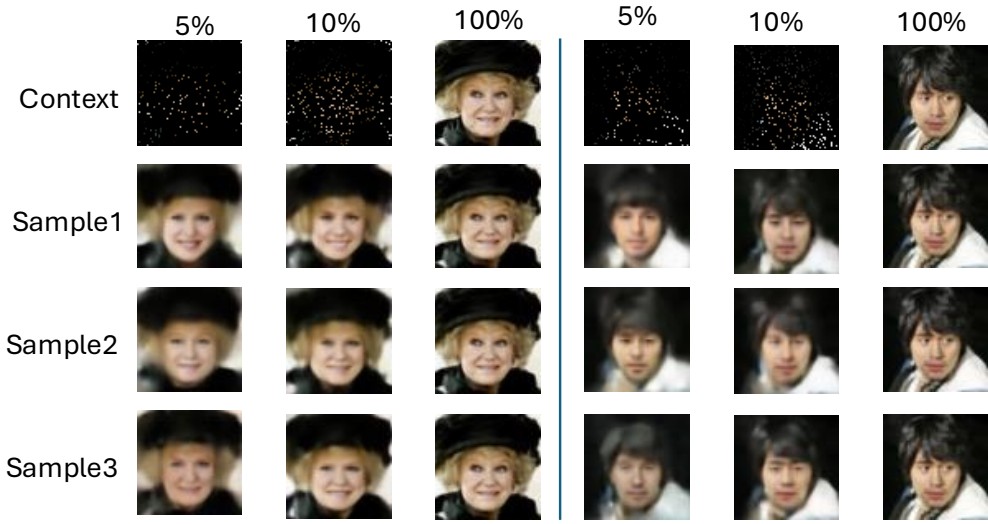

Figure 14: Diverse sampling under varying numbers of context pixels.

We also present the results with respect to different context ratio in Table 11. We report quantitative results under limited context. Specifically, we evaluate with 5%, 10%, and 50% of pixels provided as context. As shown in Table 11, performance improves as more context is available.

Table 11: PSNR versus the ratio of context pixels.

| Ratio of context pixels | PSNR |
| --- | --- |
| 5% | 22.61 |
| 10% | 24.62 |
| 50% | 28.75 |

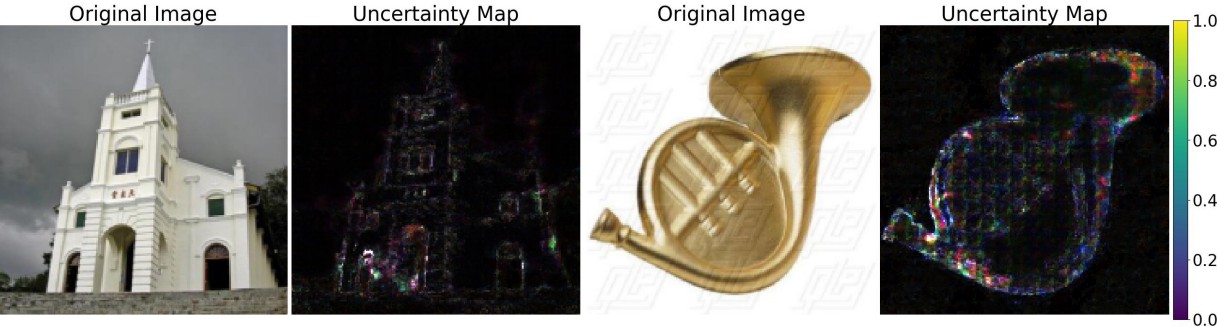

Figure 15: **Uncertainty Map of the predictions.** Edges of objects have higher uncertainty since it is more challenging for the model to capture the detailed, sharp changes at the edges. Colormap denotes per-pixel predictive variance (brighter indicates larger variance).

### F.7 Uncertainty

**Uncertainty Visualization**. As a probabilistic framework, our method can provide uncertainty estimation. To obtain the uncertainty map, we sample ten times from the predicted prior distribution to generate corresponding images and then use the variance map to represent the uncertainty. As shown in Fig. 15, high uncertainty is concentrated around the edges, which is expected, as capturing detailed, sharp changes at the edges is more challenging for the model. Additionally, we provide uncertainty maps for image completion under limited context. To demonstrate the relationship between the sparsity of the context pixels and the uncertainty, we use the distance to the nearest context pixel to quantify local sparsity (larger distances indicate sparser context). Specifically, we use quantile binning of the distance values (e.g., 12 bins) so each bin spans a similar fraction of pixels. For each bin, we report the mean uncertainty and its standard error and plot these against the bin centers, providing a stable, interpretable trend of how uncertainty evolves as locations move farther from the available context. We consider two masking schemes: random masking and lower-half masking. As shown in Fig. 16, predictive uncertainty increases with the distance to the nearest context pixel. As shown in Fig. 17, uncertainty concentrates in the masked lower half where no context pixels are provided. With sparse context, the model admits multiple plausible completions, resulting in higher predictive variance; with richer context, the model concentrates, yielding sharper, lower-variance maps that correlate with reconstruction error.

### F.8 More multi-view reconstruction results

We integrate our method into GNT (Wang et al., 2022a) framework and perform experiments on the Drums class of the NeRF synthetic dataset. Qualitative comparisons of multi-view results are presented in Fig. 18.

## G Extending G-NPF to NeRFs

**Notations.** We denote 3D world coordinates by $\mathbf{p} = (x, y, z)$ and a camera viewing direction by $\mathbf{d} = (\theta, \phi)$. Each point in 3D space have its color $\mathbf{c}(\mathbf{p}, \mathbf{d})$, which depends on the location $\mathbf{p}$ and viewing direction $\mathbf{d}$. Points also have a density value $\sigma(\mathbf{p})$ that encodes opacity. We represent coordinates and view direction together as $\mathbf{x} = \{\mathbf{p}, \mathbf{d}\}$, color and density together as $\mathbf{y}(\mathbf{p}, \mathbf{d}) = \{\mathbf{c}(\mathbf{p}, \mathbf{d}), \sigma(\mathbf{p})\}$. When observing a 3D object from multiple locations, we denote all 3D points as $\mathbf{X} = \{\mathbf{x}_n\}_{n=1}^N$ and their colors and densities as $\mathbf{Y} = \{\mathbf{y}_n\}_{n=1}^N$. Assuming a ray $\mathbf{r} = (\mathbf{o}, \mathbf{d})$ starting from the camera origin $\mathbf{o}$ and along direction $\mathbf{d}$, we sample $P$ points along the ray, with $\mathbf{x}^{\mathbf{r}} = \{x_i^{\mathbf{r}}\}_{i=1}^P$ and corresponding colors and densities $\mathbf{y}^{\mathbf{r}} = \{y_i^{\mathbf{r}}\}_{i=1}^P$. Further, we denote the observations $\widetilde{\mathbf{X}}$ and $\widetilde{\mathbf{Y}}$ as: the set of camera rays $\widetilde{\mathbf{X}} = \{\widetilde{\mathbf{x}}_n = \mathbf{r}_n\}_{n=1}^N$ and the projected 2D pixels from the rays $\widetilde{\mathbf{Y}} = \{\widetilde{\mathbf{y}}_n\}_{n=1}^N$.

**Background on Neural Radiance Fields.** We formally describe Neural Radiance Field (NeRF) (Mildenhall et al., 2021; Arandjelović & Zisserman, 2021) as a continuous function $f_{\mathrm{NeRF}} : \mathbf{x} \mapsto \mathbf{y}$, which maps 3D world coordinates $\mathbf{p}$ and viewing directions $\mathbf{d}$ to color and density values $\mathbf{y}$. That is, a NeRF function, $f_{\mathrm{NeRF}}$, is a

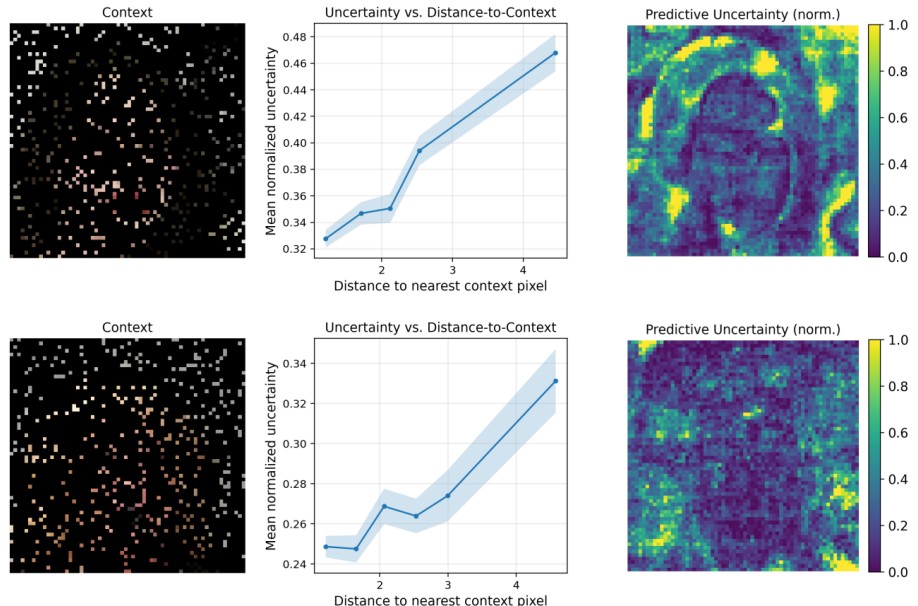

Figure 16: **Uncertainty maps for image completion (random masking).** Shown are the context image (left), the uncertainty–distance to nearest context curve (middle), and the uncertainty map (right). Local sparsity is quantified by the distance to the nearest context pixel (larger distances imply sparser context). To relate sparsity to uncertainty, we quantile-bin the distance values (e.g., 12 equal-mass bins) and plot the bin-wise mean predictive uncertainty with standard errors. Predictive uncertainty increases with distance to the nearest context pixel. Colormap denotes per-pixel predictive variance (brighter indicates larger variance).

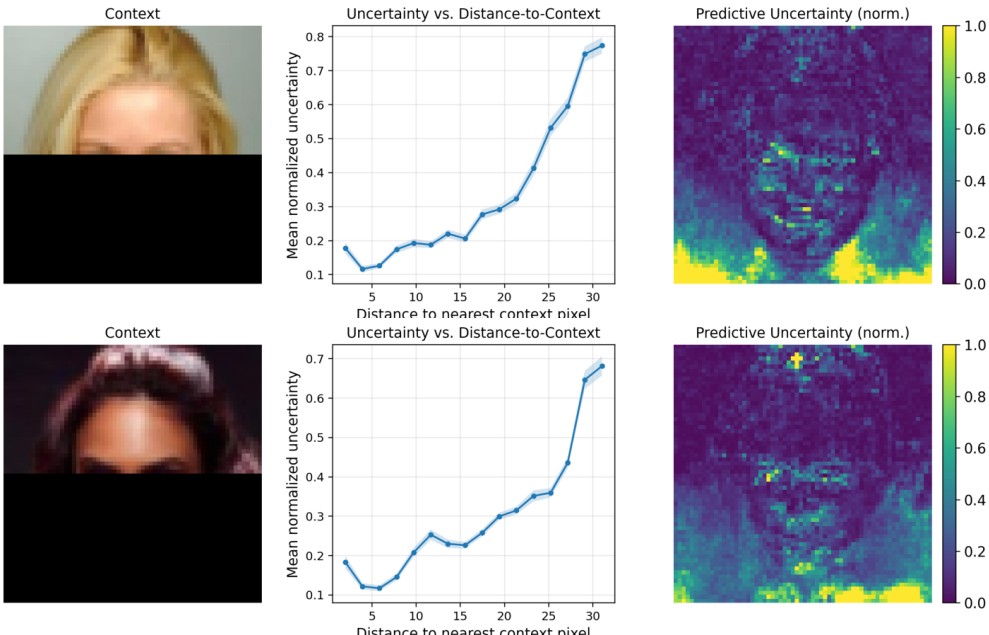

Figure 17: **Uncertainty maps for image completion (lower-half masked).** Uncertainty concentrates in the lower half where no context pixels are given. Colormap denotes per-pixel predictive variance (brighter indicates larger variance).

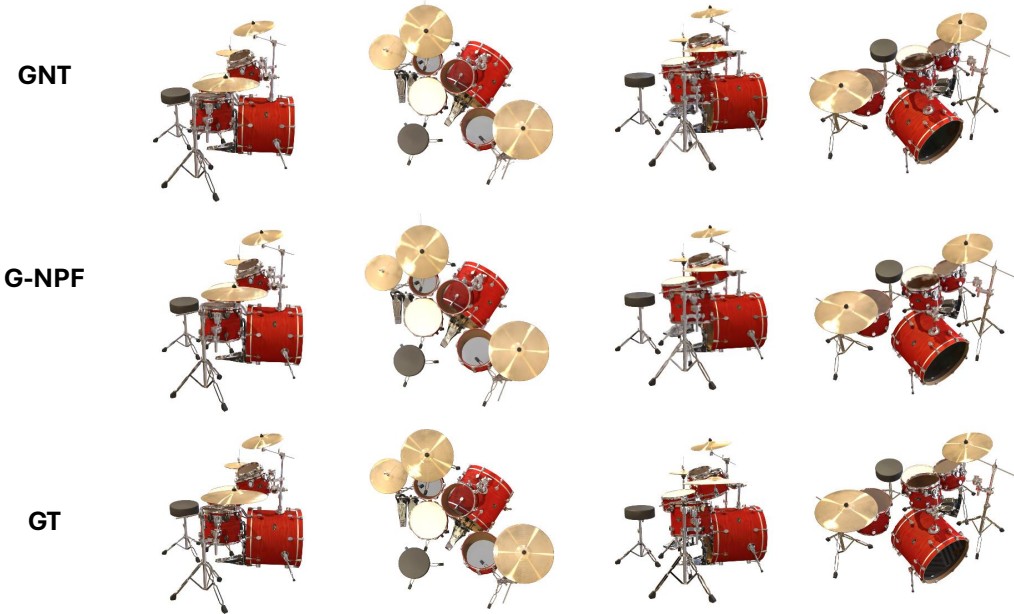

Figure 18: **Qualitative comparisons of Multi-view results on the Drums class of the NeRF synthetic dataset.**

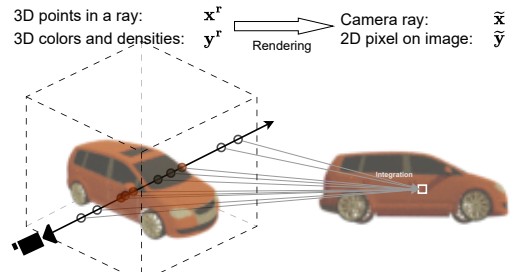

Figure 19: **Complete rendering from 3D points to a 2D pixel.**

neural network-based function that represents the whole 3D object (e.g., a car in Fig. 19) as coordinates to color and density mappings. Learning a NeRF function of a 3D object is an inverse problem where we only have indirect observations of arbitrary 2D views of the 3D object, and we want to infer the entire 3D object's geometry and appearance. With the NeRF function, given any camera pose, we can render a view on the corresponding 2D image plane by marching rays and using the corresponding colors and densities at the 3D points along the rays. Specifically, given a set of rays $\mathbf{r}$ with view directions $\mathbf{d}$, we obtain a corresponding 2D image. The integration along each ray corresponds to a specific pixel on the 2D image using the volume rendering technique described in Kajiya & Von Herzen (1984), which is also illustrated in Fig. 19. Details about the integration are given in Appendix B.

### G.1 Probabilistic NeRF Generalization

**Deterministic Neural Radiance Fields** Neural Radiance Fields are normally considered as an optimization routine in a deterministic setting (Mildenhall et al., 2021; Barron et al., 2021), whereby the function $f_{\mathrm{NeRF}}$ fits specifically to the available observations (akin to "overfitting" training data).

**Probabilistic Neural Radiance Fields** As we are not just interested in fitting a single and specific 3D object but want to learn how to infer the Neural Radiance Field of any 3D object, we focus on probabilistic Neural Radiance Fields with the following factorization:

$$p(\widetilde{\mathbf{Y}}|\widetilde{\mathbf{X}}) \propto \underbrace{p(\widetilde{\mathbf{Y}}|\mathbf{Y}, \mathbf{X})}_{\text{Integration}} \underbrace{p(\mathbf{Y}|\mathbf{X})}_{\text{NeRF Model}} \underbrace{p(\mathbf{X}|\widetilde{\mathbf{X}})}_{\text{Sampling}}. \tag{21}$$

The generation process of this probabilistic formulation is as follows. We first start from (or sample) a set of rays $\widetilde{\mathbf{X}}$. Conditioning on these rays, we sample 3D points in space $\mathbf{X}|\widetilde{\mathbf{X}}$. Then, we map these 3D points into their colors and density values with the NeRF function, $\mathbf{Y} = f_{\mathrm{NeRF}}(\mathbf{X})$. Last, we sample the 2D pixels of the viewing image that corresponds to the 3D ray $\widetilde{\mathbf{Y}}|\mathbf{Y}, \mathbf{X}$ with a probabilistic process. This corresponds to integrating colors and densities $\mathbf{Y}$ along the ray on locations $\mathbf{X}$.

The probabilistic model in Eq. (21) is for a single 3D object, thus requiring optimizing a function $f_{\mathrm{NeRF}}$ afresh for every new object, which is time-consuming. For NeRF generalization, we accelerate learning and improve generalization by amortizing the probabilistic model over multiple objects, obtaining per-object reconstructions by conditioning on context sets $\widetilde{\mathbf{X}}_C, \widetilde{\mathbf{Y}}_C$. For clarity, we use $(\cdot)_C$ to indicate context sets with a few new observations for a new object, while $(\cdot)_T$ indicates target sets containing 3D points or camera rays from novel views of the same object. Thus, we formulate a probabilistic NeRF for generalization as:

$$p(\widetilde{\mathbf{Y}}_T|\widetilde{\mathbf{X}}_T, \widetilde{\mathbf{X}}_C, \widetilde{\mathbf{Y}}_C) \propto$$
$$\underbrace{p(\widetilde{\mathbf{Y}}_T|\mathbf{Y}_T, \mathbf{X}_T)}_{\text{Integration}} \underbrace{p(\mathbf{Y}_T|\mathbf{X}_T, \widetilde{\mathbf{X}}_C, \widetilde{\mathbf{Y}}_C)}_{\text{NeRF Generalization}} \underbrace{p(\mathbf{X}_T|\widetilde{\mathbf{X}}_T)}_{\text{Sampling}}. \tag{22}$$

As this paper focuses on generalization with new 3D objects, we keep the same sampling and integrating processes as in Eq. (21). We turn our attention to the modeling of the predictive distribution $p(\mathbf{Y}_T|\mathbf{X}_T, \widetilde{\mathbf{X}}_C, \widetilde{\mathbf{Y}}_C)$ in the generalization step, which implies inferring the NeRF function.

**Misalignment between 2D context and 3D structures** It is worth mentioning that the predictive distribution in 3D space is conditioned on 2D context pixels with their ray $\{\widetilde{\mathbf{X}}_C, \widetilde{\mathbf{Y}}_C\}$ and 3D target points $\mathbf{X}_T$, which is challenging due to potential information misalignment. Thus, we need strong inductive biases with 3D structure information to ensure that 2D and 3D conditional information is fused reliably.

### G.2 Geometric Bases

To mitigate the information misalignment between 2D context views and 3D target points, we introduce geometric bases $\mathbf{B}_C = \{\mathbf{b}_i\}_{i=1}^M$, which induces prior structure to the context set $\{\widetilde{\mathbf{X}}_C, \widetilde{\mathbf{Y}}_C\}$ geometrically. $M$ is the number of geometric bases.

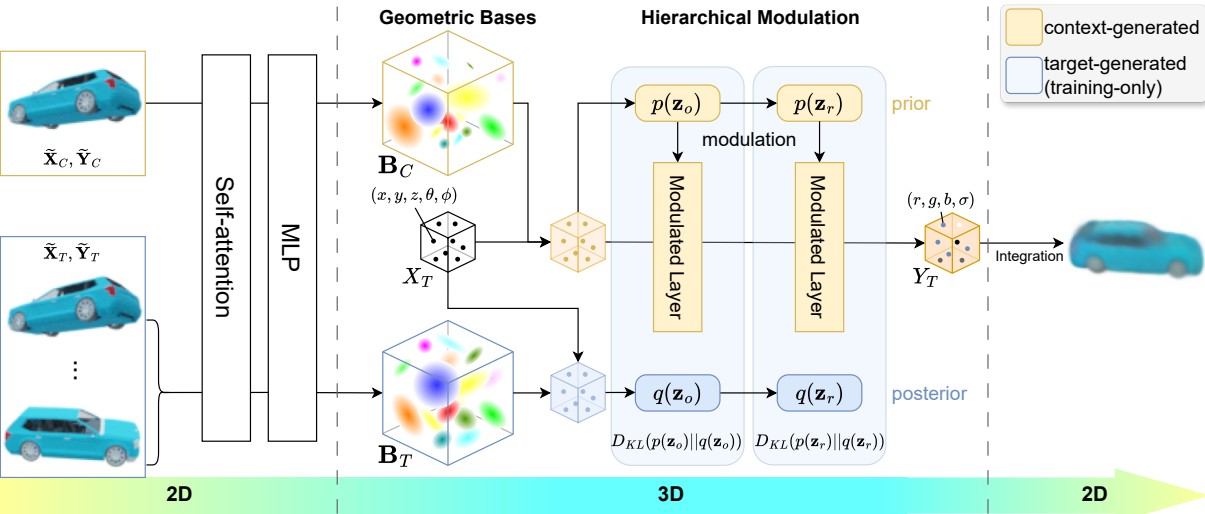

Figure 20: **Illustration of our Geometric Neural Processes.** We cast radiance field generalization as a probabilistic modeling problem. Specifically, we first construct geometric bases $\mathbf{B}_C$ in 3D space from the 2D context sets $\widetilde{\mathbf{X}}_C, \widetilde{\mathbf{Y}}_C$ to model the 3D NeRF function (Section G.2). We then infer the NeRF function by modulating a shared MLP through hierarchical latent variables $\mathbf{z}_o, \mathbf{z}_r$ and make predictions by the modulated MLP (Section G.3). The posterior distributions of the latent variables are inferred from the target sets $\widetilde{\mathbf{X}}_T, \widetilde{\mathbf{Y}}_T$, which supervises the priors during training (Section G.4).

Each geometric basis consists of a Gaussian distribution in the 3D point space and a semantic representation, *i.e.,* $\mathbf{b}_i = \{\mathcal{N}(\mu_i, \Sigma_i); \omega_i\}$, where $\mu_i$ and $\Sigma_i$ are the mean and covariance matrix of $i$-th Gaussian in 3D space, and $\omega_i$ is its corresponding latent representation. Intuitively, the mixture of all 3D Gaussian distributions implies the structure of the object, while $\omega_i$ stores the corresponding semantic information. In practice, we use a transformer-based encoder to learn the Gaussian distributions and representations from the context sets, *i.e.,* $\{(\mu_i, \Sigma_i, \omega_i)\} = \texttt{Encoder}[\widetilde{\mathbf{X}}_C, \widetilde{\mathbf{Y}}_C]$. Detailed architecture of the encoder is provided in Appendix D.1.

With the geometric bases $\mathbf{B}_C$, we review the predictive distribution from $p(\mathbf{Y}_T|\mathbf{X}_T, \widetilde{\mathbf{X}}_C, \widetilde{\mathbf{Y}}_C)$ to $p(\mathbf{Y}_T|\mathbf{X}_T, \mathbf{B}_C)$. By inferring the function distribution $p(f_{\mathrm{NeRF}})$, we reformulate the predictive distribution as:

$$p(\mathbf{Y}_T|\mathbf{X}_T, \mathbf{B}_C) = \int p(\mathbf{Y}_T|f_{\mathrm{NeRF}}, \mathbf{X}_T)p(f_{\mathrm{NeRF}}|\mathbf{X}_T, \mathbf{B}_C)df_{\mathrm{NeRF}}, \tag{23}$$

where $p(f_{\mathrm{NeRF}}|\mathbf{X}_T, \mathbf{B}_C)$ is the prior distribution of the NeRF function, and $p(\mathbf{Y}_T|f_{\mathrm{NeRF}}, \mathbf{X}_T)$ is the likelihood term. Note that the prior distribution of the NeRF function is conditioned on the target points $\mathbf{X}_T$ and the geometric bases $\mathbf{B}_C$. Thus, the prior distribution is data-dependent on the target inputs, yielding a better generalization on novel target views of new objects. Moreover, since $\mathbf{B}_C$ is constructed with continuous Gaussian distributions in the 3D space, the geometric bases can enrich the locality and semantic information of each discrete target point, enhancing the capture of high-frequency details (Chen et al., 2023b; 2022; Müller et al., 2022).

### G.3 Geometric Neural Processes with Hierarchical Latent Variables

With the geometric bases, we propose Geometric Neural Processes (**G-NPF**) by inferring the NeRF function distribution $p(f_{\mathrm{NeRF}}|\mathbf{X}_T, \mathbf{B}_C)$ in a probabilistic way. Based on the probabilistic NeRF generalization in Eq. (22), we introduce hierarchical latent variables to encode various spatial-specific information into $p(f_{\mathrm{NeRF}}|\mathbf{X}_T, \mathbf{B}_C)$, improving the generalization ability in different spatial levels. Since all rays are independent of each other, we decompose the predictive distribution in Eq. (23) as:

$$p(\mathbf{Y}_T|\mathbf{X}_T, \mathbf{B}_C) = \prod_{n=1}^{N} p(\mathbf{y}_T^{\mathbf{r},n}|\mathbf{x}_T^{\mathbf{r},n}, \mathbf{B}_C), \tag{24}$$

where the target input $\mathbf{X}_T$ consists of $N \times P$ location points $\{\mathbf{x}_T^{\mathbf{r},n}\}_{n=1}^N$ for $N$ rays.

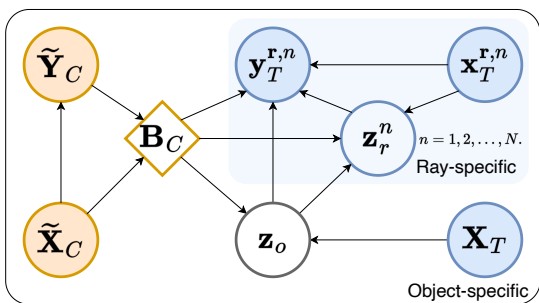

Figure 21: **Graphical model for the proposed geometric neural processes.**

Further, we develop a hierarchical Bayes framework for G-*NPF* to accommodate the data structure of the target input $\mathbf{X}_T$ in Eq. (24). We introduce an object-specific latent variable $\mathbf{z}_o$ and $N$ individual ray-specific latent variables $\{\mathbf{z}_r^n\}_{n=1}^N$ to represent the randomness of $f_{\mathrm{NeRF}}$.

Within the hierarchical Bayes framework, $\mathbf{z}_o$ encodes the entire object information from all target inputs and the geometric bases $\{\mathbf{X}_T, \mathbf{B}_C\}$ in the global level; while every $\mathbf{z}_r^n$ encodes ray-specific information from $\{\mathbf{x}_T^{\mathbf{r},n}, \mathbf{B}_C\}$ in the local level, which is also conditioned on the global latent variable $\mathbf{z}_o$. The hierarchical architecture allows the model to exploit the structure information from the geometric bases $\mathbf{B}_C$ in different levels, improving the model's expressiveness ability. By introducing the hierarchical latent variables in Eq. (24), we model G-*NPF* as:

$$p(\mathbf{Y}_T|\mathbf{X}_T, \mathbf{B}_C) = \int \prod_{n=1}^N \Big\{ \int p(\mathbf{y}_T^{\mathbf{r},n}|\mathbf{x}_T^{\mathbf{r},n}, \mathbf{B}_C, \mathbf{z}_r^n, \mathbf{z}_o)$$
$$p(\mathbf{z}_r^n|\mathbf{z}_o, \mathbf{x}_T^{\mathbf{r},n}, \mathbf{B}_C) d\mathbf{z}_r^n \Big\} p(\mathbf{z}_o|\mathbf{X}_T, \mathbf{B}_C) d\mathbf{z}_o, \tag{25}$$

where $p(\mathbf{y}_T^{\mathbf{r},n}|\mathbf{x}_T^{\mathbf{r},n}, \mathbf{B}_C, \mathbf{z}_o, \mathbf{z}_r^i)$ denotes the ray-specific likelihood term. In this term, we use the hierarchical latent variables $\{\mathbf{z}_o, \mathbf{z}_r^i\}$ to modulate a ray-specific NeRF function $f_{\mathrm{NeRF}}$ for prediction, as shown in Fig. 20. Hence, $f_{\mathrm{NeRF}}$ can explore global information of the entire object and local information of each specific ray, leading to better generalization ability on new scenes and new views. A graphical model of our method is provided in Fig. 21.

In the modeling of G-*NPF*, the prior distribution of each hierarchical latent variable is conditioned on the geometric bases and target input. We first represent each target location by integrating the geometric bases, *i.e.*, $< \mathbf{x}_T^n, \mathbf{B}_C >$, which aggregates the relevant locality and semantic information for the given input. Since $\mathbf{B}_C$ contains $M$ Gaussians, we employ a Gaussian radial basis function in Eq. (26) between each target input $\mathbf{x}_T^n$ and each geometric basis $\mathbf{b}_i$ to aggregate the structural and semantic information to the 3D location representation. Thus, we obtain the 3D location representation as follows:

$$< \mathbf{x}_T^n, \mathbf{B}_C > = \mathrm{MLP}\Big[ \sum_i^M \exp(-\frac{1}{2}(\mathbf{x}_T^n - \mu_i)^T \Sigma_i^{-1}(\mathbf{x}_T^n - \mu_i)) \cdot \omega_i \Big], \tag{26}$$

where $\mathrm{MLP}[\cdot]$ is a learnable neural network. With the location representation $< \mathbf{x}_T^n, \mathbf{B}_C >$, we next infer each latent variable hierarchically, in object and ray levels.

**Object-specific Latent Variable.** The distribution of the object-specific latent variable $\mathbf{z}_o$ is obtained by aggregating all location representations:

$$[\mu_o, \sigma_o] = \mathrm{MLP}\Big[ \frac{1}{N \times P} \sum_{n=1}^N \sum_{\mathbf{r}} < \mathbf{x}_T^n, \mathbf{B}_C > \Big], \tag{27}$$

where we assume $p(\mathbf{z}_o|\mathbf{B}_C, \mathbf{X}_T)$ is a standard Gaussian distribution and generate its mean $\mu_o$ and variance $\sigma_o$ by a $\mathrm{MLP}$. Thus, our model captures objective-specific uncertainty in the NeRF function.

**Ray-specific Latent Variable.** To generate the distribution of the ray-specific latent variable, we first average the location representations ray-wisely. We then obtain the ray-specific latent variable by aggregating the averaged location representation and the object latent variable through a lightweight transformer. We formulate the inference of the ray-specific latent variable as:

$$[\mu_r, \sigma_r] = \texttt{Transformer}\Big[\texttt{MLP}[\frac{1}{P}\sum_{\mathbf{r}} <\mathbf{x}_T^n, \mathbf{B}_C>]; \hat{\mathbf{z}}_o\Big], \tag{28}$$

where $\hat{\mathbf{z}}_o$ is a sample from the prior distribution $p(\mathbf{z}_o|\mathbf{X}_T, \mathbf{B}_C)$. Similar to the object-specific latent variable, we also assume the distribution $p(\mathbf{z}_r^n|\mathbf{z}_o, \mathbf{x}_T^{\mathbf{r},n}, \mathbf{B}_C)$ is a mean-field Gaussian distribution with the mean $\mu_r$ and variance $\sigma_r$. We provide more details of the latent variables in Appendix D.2.

**NeRF Function Modulation.** With the hierarchical latent variables $\{\mathbf{z}_o, \mathbf{z}_r^n\}$, we modulate a neural network for a 3D object in both object-specific and ray-specific levels. Specifically, the modulation of each layer is achieved by scaling its weight matrix with a style vector (Guo et al., 2023). The object-specific latent variable $\mathbf{z}_o$ and ray-specific latent variable $\mathbf{z}_r^n$ are taken as style vectors of the low-level layers and high-level layers, respectively. The prediction distribution $p(\mathbf{Y}_T|\mathbf{X}_T, \mathbf{B}_C)$ are finally obtained by passing each location representation through the modulated neural network for the NeRF function. More details are provided in Appendix D.3.

### G.4  Empirical Objective

**Evidence Lower Bound.** To optimize the proposed G-*NPF*, we apply variational inference (Garnelo et al., 2018b) and derive the evidence lower bound (ELBO) as:

$$\log p(\mathbf{Y}_T|\mathbf{X}_T, \mathbf{B}_C) \geq$$
$$\mathbb{E}_{q(\mathbf{z}_o|\mathbf{B}_T, \mathbf{X}_T)}\Big\{ \sum_{n=1}^{N} \mathbb{E}_{q(\mathbf{z}_r^n|\mathbf{z}_o, \mathbf{x}_T^{\mathbf{r},n}, \mathbf{B_T})} \log p(\mathbf{y}_T^{\mathbf{r},n}|\mathbf{x}_T^{\mathbf{r},n}, \mathbf{z}_o, \mathbf{z}_r^n)$$
$$- D_{\mathrm{KL}}[q(\mathbf{z}_r^n|\mathbf{z}_o, \mathbf{x}_T^{\mathbf{r},n}, \mathbf{B}_T)||p(\mathbf{z}_r^n|\mathbf{z}_o, \mathbf{x}_T^{\mathbf{r},n}, \mathbf{B}_C)]\Big\}$$
$$- D_{\mathrm{KL}}[q(\mathbf{z}_o|\mathbf{B}_T, \mathbf{X}_T)||p(\mathbf{z}_o|\mathbf{B}_C, \mathbf{X}_T)], \tag{29}$$

where $q_{\theta,\phi}(\mathbf{z}_o, \{\mathbf{z}_r^i\}_{i=1}^N|\mathbf{X}_T, \mathbf{B}_T) = \Pi_{i=1}^N q(\mathbf{z}_r^n|\mathbf{z}_o, \mathbf{x}_T^{\mathbf{r},n}, \mathbf{B}_T)q(\mathbf{z}_o|\mathbf{B}_T, \mathbf{X}_T)$ is the involved variational posterior for the hierarchical latent variables. $\mathbf{B}_T$ is the geometric bases constructed from the target sets $\{\widetilde{\mathbf{X}}_T, \widetilde{\mathbf{Y}}_T\}$, which are only accessible during training. The variational posteriors are inferred from the target sets during training, which introduces more information on the object. The prior distributions are supervised by the variational posterior using Kullback–Leibler (KL) divergence, learning to model more object information with limited context data and generalize to new scenes. Detailed derivations are provided in Appendix **??**.

For the geometric bases $\mathbf{B}_C$, we regularize the spatial shape of the context geometric bases to be closer to that of the target one $\mathbf{B}_T$ by introducing a KL divergence. Therefore, given the above ELBO, our objective function consists of three parts: a reconstruction loss (MSE loss), KL divergences for hierarchical latent variables, and a KL divergence for the geometric bases. The empirical objective for the proposed G-*NPF* is formulated as:

$$\mathcal{L}_{\text{G-}NPF} = ||y - y'||_2^2 + \alpha \cdot \big(D_{\mathrm{KL}}[p(\mathbf{z}_o|\mathbf{B}_C)|q(\mathbf{z}_o|\mathbf{B}_T)]$$
$$+ D_{\mathrm{KL}}[p(\mathbf{z}_r|\mathbf{z}_o, \mathbf{B}_C)|q(\mathbf{z}_r|\mathbf{z}_o, \mathbf{B}_T)]\big) + \beta \cdot D_{\mathrm{KL}}[\mathbf{B}_C, \mathbf{B}_T], \tag{30}$$

where $y'$ is the prediction. $\alpha$ and $\beta$ are hyperparameters to balance the three parts of the objective. The KL divergence on $\mathbf{B}_C, \mathbf{B}_T$ is to align the spatial location and the shape of two sets of bases.

## H   More Related Work

**Generalizable Neural Radiance Fields (NeRF)**   Advancements in neural radiance fields have focused on improving generalization across diverse scenes and objects. Wang et al. (2022a) propose an attention-based NeRF architecture, demonstrating enhanced capabilities in capturing complex scene geometries by

focusing on informative regions. Suhail et al. (2022) introduce a generalizable patch-based neural rendering approach, enabling models to adapt to new scenes without retraining. Xu et al. (2022) present *Point-NeRF*, leveraging point-based representations for efficient scene modeling and scalability. Wang et al. (2024) further enhance point-based methods by incorporating visibility and feature augmentation to improve robustness and generalization. Liu et al. (2024) propose a geometry-aware reconstruction with fusion-refined rendering for generalizable NeRFs, improving geometric consistency and visual fidelity. Recently, the *Large Reconstruction Model (LRM)* (Hong et al., 2023) has drawn attention. It aims for single-image to 3D reconstruction, emphasizing scalability and handling of large datasets.

**Gaussian Splatting-based Methods**  Gaussian splatting (Kerbl et al., 2023) has emerged as an effective technique for efficient 3D reconstruction from sparse views. Szymanowicz et al. (2024) propose *Splatter Image* for ultra-fast single-view 3D reconstruction. Charatan et al. (2024) introduce *pixelsplat*, utilizing 3D Gaussian splats from image pairs for scalable generalizable reconstruction. Chen et al. (2025) present *MVSplat*, focusing on efficient Gaussian splatting from sparse multi-view images. Our approach can be a complementary module for these methods by introducing a probabilistic neural processing scheme to fully leverage the observation.

**Diffusion-based 3D Reconstruction**  Integrating diffusion models into 3D reconstruction has shown promise in handling uncertainty and generating high-quality results. Müller et al. (2023) introduce *DiffRF*, a rendering-guided diffusion model for 3D radiance fields. Tewari et al. (2023) explore solving stochastic inverse problems without direct supervision using diffusion with forward models. Liu et al. (2023) propose *Zero-1-to-3*, a zero-shot method for generating 3D objects from a single image without training on 3D data, utilizing diffusion models. Shi et al. (2023a) introduce *Zero123++*, generating consistent multi-view images from a single input image using diffusion-based techniques. Shi et al. (2023c) present *MVDream*, which uses multi-view diffusion for 3D generation, enhancing the consistency and quality of reconstructed models.

