# OpenReview forum: "Geometric Neural Process Fields"
_TMLR — Decision pending for TMLR_

### Review · Reviewer_ueCt · 2025-07-22

**Summary Of Contributions:**

This paper presents an extension of Neural Processes [Garnelo 2018] tailored specifically for neural fields representing geometric structures (1D / 2D / 3D). It aims to perform probabilistic inference of neural fields (represented in function space) conditioned on a set of points and function values (for 1D and 2D signals), or indirect/partial observations of them (for 3D rendered NeRFs). It defines a generative model that incorporates a local basis-function representation for latent variables (where those basis functions are placed in the domain of functions to be represented, and allow spatially-local features), and an encoder incorporating self-attention. Experiments are conducted on time-series, images, and 3D scenes, considering direct reconstruction/regression, completion from sparse data, and 3D reconstruction given posed images.

**Audience:**

Yes

**Broader Impact Concerns:**

This work does not raise any significant concerns.

**Claims And Evidence:**

No

**Requested Changes:**

It is not explicitly stated how z is marginalised during inference – presumably it is using some finite set of samples from the posterior $q(z |  x, y)$? This should be mentioned somewhere.

Provide quantitative and qualitative results showing diversity of samples. It is unclear currently what complexity of posterior dependencies can be captured by the method. The global latent should presumably be able to induce fairly arbitrary dependencies among the local ones, but does it actually learn to do this, as opposed to simply packing information into the local variables and ignoring the global one? This should be apparent by seeing diverse yet coherent global structures in qualitative results.

Discuss what the neural process is expected to learn from the single-scene NeRF-synthetic experiments, where there is only a single function to learn.

Add some discussion of the probabilistic NeRF reconstruction methods mentioned above, and preferably compare against at least one of them.

Sec 3.2 says typically unstructured MLPs are used; however most practical generative models over NeRFs do in fact use more structured representations (for example EG3D's triplanes, DDIBR's image-supported feature grids, etc.). This should be clarified.

Sec 3.2 also mentions 'smaller training datasets'; however it is unclear how this is justified, and it is not followed up anywhere. Please either expand this point, or remove it. Currently it is not even clear whether it refers to 'meta datasets' (i.e. dataset of scenes / images) or 'inner' datasets (e.g. pixels within an image).

Clarify why the uncertainty plot looks as it does (see mention under weaknesses). Also add uncertainty and diversity results for the other tasks (completion and reconstruction) where there is actually some ambiguity to be represented.

While the geometric basis functions are a natural approach, is it not true that for image regression, they mean the task at hand (i.e. autoencoding) reduces to the model directly copying from each input pixel to the corresponding output basis function weight?

What does "suffer from the unconstrained nature of stochastic gradient descent" on p3 mean, how is it relevant here, and does the proposed method address this?

Typo at start app D.3: "We use modulation to The latent variables for modulating"

**Strengths And Weaknesses:**

**Strengths**

The proposed model is novel, with a logical extension to the original Neural Processes to better support neural fields intended to represent geometric / spatial structures, as well as a more modern encoder architecture.

The method is quite general, and experiments are conducted on several different domains and tasks to demonstrate this – 1D (time series), 2D (images – autoencoding and completion), 3D (novel view synthesis).

Qualitative and quantitative (measured by PSNR) results on autoencoding 2D images are visually good, and exceed the performance of the respective baselines. Where standard deviations on metrics are given, the differences appear to be significant.

Results on novel view synthesis for ShapeNet renderings results slightly exceed the selected baselines, yielding plausible shapes under significant changes of viewpoint, given just one or two input camera views. Quantitatively, the results exceed NeRF-VAE and some other recent approaches to neural processes.

The paper is clear, well-organised and pleasant to read throughout.

**Weaknesses**

The 3D experiments are only conducted on ShapeNet, which is very simplistic compared with other modern works on probabilistic 3D reconstruction, such as Viewset Diffusion [Szymanowicz, ICCV 2023] and Denoising Diffusion via Image-Based Rendering [Anciukevicius, ICLR 2024]. This makes it difficult to understand how the method compares with recent approaches.

Section 3 says generalisation in NeRFs is challenging, and gives very few references, implying this is a little-studied problem; however this is somewhat misleading given the rather large literature on this area. While there is some additional discussion of these, and of probabilistic NeRFs, in App. H, this is rather brief, and should be promoted to the main text at least in part. Even the seminal pixel-NeRF [Yu, CVPR 2021] is missing from discussion.

For the generative case, notably Viewset Diffusion [Szymanowicz, ICCV 2023] and Denoising Diffusion via Image-Based Rendering [Anciukevicius, ICLR 2024] should be discussed and compared against; these perform probabilistic reconstruction of neural fields given sparse images, and effectively have latent variables supported on rays cast from multi-view images, hence are clearly very relevant.

The image regression and ShapeNet reconstruction results give only PSNR as the metric. This is sub-optimal since it correlates poorly with perceptual similarity; it would be valuable to also include LPIPS and SSIM (as for the experiments on NeRF-synthetic).

Given that the method is probabilistic, there is insufficient evidence of sample diversity. In particular, the only relevant visualisation is for the 2D image regression task, where higher variability is shown to arise at sharp boundaries in pixel space.

Instead, to demonstrate the model is behaving probabilistically and sampling a reasonable distribution, diverse samples should be given for few inputs, both for the image-completion and 3D reconstruction tasks. These should show the model can generate diverse plausible samples that are compatible with the input conditioning (e.g. plausible completions for the back of an object when imaging only the front).

Moreover, the visualisation of higher variance at edges does not seem to be showing anything meaningful. Here the model is conditioned on a complete image and tasked with reconstructing is; there is *zero* predictive uncertainty anywhere in the reconstruction, and the fact that variability is higher at the edges seems to be a (minor) failure mode of the model rather than a demonstration of correct sampling behavior.

The image completion task lacks quantitative results, and baselines. Instead only a single figure with qualitative results from the proposed method is given. It would also be good to see results with even fewer context pixels, as in the original Neural Processes paper.

For the NeRF-VAE baseline, quantitative results are only given for one view not two, despite NeRF-VAE supporting varying numbers of context views.

The text seems to imply the experiments on NeRF-synthetic dataset use per-scene models, instead of 'global' models encompassing the entire dataset (and this seems inevitable given the small size of that dataset meaning generalisation is impractical). If this is the case, this experiment does not seem to be appropriate – neural processes are intended to learn a distribution over functions given a 'meta-dataset' of sets of samples from instantiations of functions from the underlying distribution; here however there is only a single set of points, and a single function to be found.

---

> ### Author Response · Authors · 2025-10-16
>
> *We sincerely thank # Reviewer ueCt for their insightful comments. The following addresses their concerns and provides answers to their questions. All revisions in the manuscript are highlighted in blue.*
>
> ---
>
> **W1 & Requested changes (RC) 4. Comparison with Viewset Diffusion [Szymanowicz, ICCV 2023] and Denoising Diffusion via Image-Based Rendering [Anciukevicius, ICLR 2024].**
>
> Thank you for pointing to Viewset Diffusion (Szymanowicz, ICCV 2023) and DD-IBR (Anciukevicius, ICLR 2024). We have added the discussion in the revised version in Section 5 and experimental comparison in Section F.3.
>
> Both Viewset Diffusion and  DD-IBR are generative diffusion models that learn data-level score priors (over viewsets/IB-planes) and adapt to a new scene via diffusion process; their ability to generalize across scenes largely follows from how well the learned prior covers the training distribution, with per-scene adaptation expressed through new sampling trajectories. In contrast, our approach is meta-learning in the Neural-Process paradigm: during training, we learn a task-level prior over functions (neural fields) across many scenes, and at test time we infer the task function for reconstruction.
> This yields few-shot generalization across scenes with constant-time per-scene adaptation, favorable scalability when processing many new scenes, and robustness to view/resolution changes due to operating directly in function space—complementary to diffusion’s strengths as a powerful generative prior.
>
>
> As Viewset Diffusion (Szymanowicz et al., ICCV 2023) is the only method with public code, we evaluate on the ShapeNet Car subset and reuse its default encoder architecture. Specifically, we adopt its encoder as the backbone to estimate our geometric bases and then apply our hierarchical modulation for reconstruction.
>
> | Method                  | PSNR ↑  | SSIM ↑  | LPIPS ↓ |
> |--------------------------|:-------:|:-------:|:-------:|
> | Viewset Diffusion | 28.00   | 0.871   | 0.167   |
> | **Ours**                 | **28.68** | **0.892** | **0.153** |
>
> **W2: More related work discussion, including pixel-NeRF [Yu, CVPR 2021]**
>
> Thank you for the suggestions. We have moved the NeRF generalization-related works from Section H to Section 5,  in the main manuscript. We also discuss the connection with the seminal work, including PixelNerf [Yu, CVPR2021], Viewset Diffusion [Szymanowicz, ICCV 2023] and Denoising Diffusion via Image-Based Rendering [Anciukevicius, ICLR 2024].
>
>
> **W3: Add SSIM and LIPIS.**
>
> Thank you for your suggestion. We have included the SSIM and LPIPS metrics for both the image regression and ShapeNet reconstruction results. Following the implicit neural fields generalization setting (Trans-INR (Chen & Wang, 2022); VNP (Guo et al., 2023)), which do not originally report these two metrics, we computed the SSIM and LPIPS scores as follows: for 3D, we used the pretrained weights provided by VNP; for 2D, we re-ran the only available code implementation, Trans-INR, and calculated the metrics accordingly. Our method consistently outperforms the baseline on SSIM and LPIPS scores.
>
> These additional results are presented in Appendix F.1 to keep the comparisons in the main manuscript consistent with the standard benchmarks used in the baselines.
>
>
> | Car | PSNR ↑ | SSIM ↑ | LPIPS ↓ |
> |:-------|:------:|:------:|:------:|
> | VNP    | 24.21  | 0.92   | 0.13   |
> | **Ours** | **25.13** | **0.95** | **0.11** |
>
> | Lamps | PSNR ↑ | SSIM ↑ | LPIPS ↓ |
> |:-------|:------:|:------:|:------:|
> | VNP    | 24.10  | 0.91   | 0.16   |
> | **Ours** | **24.59** | **0.92** | **0.14** |
>
>
> | Chair | PSNR ↑ | SSIM ↑ | LPIPS ↓ |
> |:-------|:------:|:------:|:------:|
> | VNP    | 19.54  | 0.83   | 0.22   |
> | **Ours** | **20.74** | **0.87** | **0.19** |
>
>
> | Method | PSNR ↑ (CelebA) | SSIM ↑ (CelebA) | LPIPS ↓ (CelebA) | PSNR ↑ (Imagenette) | SSIM ↑ (Imagenette) | LPIPS ↓ (Imagenette) |
> |:--------|:---------------:|:----------------:|:----------------:|:-------------------:|:--------------------:|:--------------------:|
> | Trans-INR | 31.96 | 0.92 | 0.07 | 29.01 | 0.86 | 0.14 |
> | **Ours**  | **33.41** | **0.96** | **0.04** | **29.82** | **0.93** | **0.09** |

---

> ### Author Response · Authors · 2025-10-16
>
> **W4 & RC2: Diverse sampling from the distribution.**
>
> We would like to first clarify that Neural Processing is not a generative model, like VAE, where the primary goal is to produce diverse samples. Instead, NPs focus on efficiently leveraging available context to reduce predictive uncertainty: with limited context, they yield higher uncertainty (and thus more diverse samples), while with richer context the predictions become more certain (and less diverse). This behavior is canonical, e.g., in ANP (see Fig.~4(a)), using only 10% context pixels already leads to visually similar samples.
>
> In Appendix F.6, we provide qualitative examples and quantitative results for the image completion task where fewer context pixels are given. The results demonstrate that our method can generate outputs that are both visually similar to the ground truth and exhibit meaningful diversity across samples.
>
> | Seed | PSNR  |
> |:----:|:------:|
> | 0 | 24.97 |
> | 1 | 24.62 |
> | 2 | 24.26 |
>
>
> For the 3D reconstruction task, the difference is visually small, while the PSNR is different. This also happens for VNP (Guo et al., 2023) in this task when sampling with different random seeds. We hypothesize that this is due to the color and shape of the object are known from the given view.. Hence, we provide the quantitative results with different random seeds.
>
> | Seed | PSNR  |
> |:----:|:------:|
> | 0 | 25.01 |
> | 1 | 24.99 |
> | 2 | 25.15 |
>
>
> **W5 & RC7: Visualisation of uncertainty map.**
>
> Thank you for the comment. We would like to clarify that even when conditioning on the full image, a small amount of residual uncertainty can remain, primarily due to practical model/likelihood mismatch (e.g., discretization, anti-aliasing, small measurement noise). Such residuals naturally concentrate at high-frequency boundaries, which explains the higher variance near edges in our visualization. Even when overfitting on one image, such high-frequency details are not easy for the implicit neural function. Hence, uncertainty exists.
>
> To avoid confusion, we move this figure to Appendix F.7 and additionally provide uncertainty maps for the image-completion setting, where NPs exhibit higher uncertainty under limited context. As shown in Figure 16, uncertainty concentrates in regions with fewer context points.
>
>
> **W6 : Image completion task lacks quantitative results with fewer pixels given.**
>
> We report quantitative results under the limited context. Specifically, we evaluate with 5%, 10%, and 50% of pixels provided as context. As shown in the table below, performance improves as more context is available.
> We have added these results in Appendix F.6.
>
> | Ratio of Context Pixels | PSNR  |
> |:-----------------------:|:-----:|
> | 5%                      | 22.61 |
> | 10%                     | 24.62 |
> | 50%                     | 28.75 |
>
>
> **W7: NeRF-VAE with two views.**
>
> For the NeRF-VAE result reported in Table 2, we use the numbers from VNP (Guo et al., 2023).  NeRF-VAE does not provide its implementation. To address your concern, we reimplement NeRF-VAE and report the results for two views.
>
> | Method   | Cars  | Lamps | Chairs |
> |:---------:|:-----:|:-----:|:------:|
> | NeRF-VAE | 23.02 | 21.89 | 18.35  |
> | Ours     | 26.39 | 25.32 | 22.68  |
>
>
> **W8 & RC3: Meta-learning on NeRF-synthetic dataset.**
>
> In our NeRF-synthetic experiments, “generalization” follows the standard protocol and refers to novel-view prediction within a scene rather than across scenes. Our formulation is NP-like but targets a distribution over view-specific functions (i.e., the conditional radiance field given a context set for that scene), not a distribution over functions across a large meta-dataset of scenes. Even though cross-scene meta-generalization is not the aim of this benchmark, our probabilistic method provides better qualitative results than the deterministic baseline when training on the Drums category but testing on the LEGO category (see Figure 10).  We have clarified this in Appendix F.3.
>
> **RC1: How $z$ is marginalised for inference?**
>
> At test time we only have the context set, so we use the learned priors \\(p(z_g \\mid x_T,B_C)\\) and \\(p(z_{l,m} \\mid z_g,x_{T,m},B_C)\\) and approximate the marginalization over \\(z\\) by Monte Carlo. Concretely, we draw \\(S\\) samples of the global latent \\(z_g \\sim p(z_g \\mid x_T,B_C)\\) and, for each target coordinate \\(m\\), draw \\(T\\) samples \\(z_{l,m} \\sim p(z_{l,m} \\mid z_g,x_{T,m},B_C)\\), then average the predictive likelihoods \\(p(y_{T,m} \\mid z_g,z_{l,m},x_{T,m},B_C)\\).
> In practice, we use \\(S=T=5\\) for all quantitative metrics.
>
> We have clarified this in Section 3.4.

---

> > ### Author Response · Authors · 2025-10-16
> >
> > **RC 5&6: Sec. 3.2 Clarification: unstructured MLPs and 'smaller training datasets';**
> >
> > We apologize for the confusion. For the unstructured MLPs, we refer to the general implicit neural fields setting. We also noticed the use of the structured representation in the NeRF setting. So, we have revised the text as:
> >
> > While some NeRF pipelines introduce structured feature backbones (e.g., planes or grids), such designs are task- and modality-specific; there is no single unified structured representation that seamlessly covers heterogeneous signal types (e.g., 2D images and 3D radiance/shape fields).
> >
> > For the 'smaller training datasets', we refer to the meta dataset, so we have removed it and only keep the “few-shot learning settings”.
> >
> >
> > **RC8: geometric basis functions for image regression**
> >
> > No, it is not directly copying from each input pixel. In our formulation, the image is modeled as a continuous field:
> >
> >
> > $$\\hat{y}(\\mathbf{x}) = \\sum_{k=1}^{K} w_k \\left(\\mathbf{x}, B_C,z \\right)\\phi_k(\\mathbf{x}),$$
> >
> > where \\(\\{\\phi_k\\}\\) are geometric basis functions, and the *coefficients are not stored per pixel* but are *predicted as functions of the coordinates* \\(\\mathbf{x}\\), conditioned on the context \\(B_C\\) and latents \\(z\\).
> >
> > This rules out trivial pixel-wise copying because:  (i) parameter sharing & continuity: a single predictor produces \\(w_k(\\mathbf{x})\\) for all \\(\\mathbf{x}\\), so unseen locations must be interpolated or extrapolated by prediction rather than copied;
> > (ii) compression: we use \\(K \\ll \\#\text{pixels}\\), precluding a one-to-one pixel to weight mapping;
> > (iii) generalization: tasks like completion or novel-target prediction involve unobserved pixels that must be inferred from \\(B_C\\), again incompatible with direct copying.
> >
> > Empirically, our completion results and uncertainty maps exhibit diverse, plausible fills rather than per-pixel replication.
> >
> >
> > **RC8: "the unconstrained nature of stochastic gradient descent"**
> >
> > By “the unconstrained nature of stochastic gradient descent”, we mean that, when training over-parameterized coordinate to value MLPs with standard pixel-wise losses, SGD is free to converge to many near-interpolating solutions that fit the context pixels but behave arbitrarily between them, i.e., little functional/structural guidance prevents memorization or spurious high-frequency artifacts, which is especially harmful in few-shot regimes. Our method introduces explicit constraints that counteract this: (i) a geometric basis expansion with small K promotes spatial coherence; (ii) hierarchical, context-conditioned priors with KL regularization couple scenes and penalize per-scene drift.
> >
> > **RC10: Typo at Appendix D.3.**
> >
> > Thank you for pointing out the typo. We have fixed the typo in Appendix D.3.
> >
> > ---
> >
> > *Thank you for your time and efforts. We hope the provided visualization and clarifications have addressed your concerns. Please don’t hesitate to reach out if you have further questions or need more information.*

---

> > > ### Author Response · Authors · 2025-10-27
> > > **Update on uncertainty map**
> > >
> > > In this comment, we provide the updates regarding the uncertainty map improvement, which is related to the reviewer ueCt's concern about the uncertainty map.
> > >
> > > To better demonstrate the uncertainty map, we (a) plot predictive uncertainty versus distance to the nearest context pixel and (b) add a lower-half masked visualization.
> > >
> > > For (a), we use the distance to the nearest context pixel to quantify local sparsity (larger distances indicate sparser context), and present its relationship to predictive uncertainty. Specifically, we use quantile binning of the distance values (e.g., 12 bins) so each bin spans a similar fraction of pixels rather than a fixed numeric range. For each bin, we report the mean uncertainty and its standard error and plot these against the bin centers, providing a stable, interpretable trend of how uncertainty evolves as locations move farther from the available context. As shown in Fig.16, uncertainty increases with distance to the nearest context pixel.
> > >
> > >  For (b), Fig. 17 shows higher uncertainty concentrating in the masked lower half where no context pixels are provided. Most pixels follow the same conclusion that uncertainty increases with distance to the nearest context pixel.

---

### Review · Reviewer_snNU · 2025-08-18

**Summary Of Contributions:**

The paper tackles neural field generalization by extending Neural Processes (NP) with geometric and hierarchical priors. The authors propose G-NPF, which adapts to new scenes with limited observations while following a probabilistic framework and exploiting geometric priors.

**Core Framework:** Building on the standard NP formulation $p(yT|xT,C)=∫p(yT|xT,z)p(z|C)dz$,
they formulate neural field inference as probabilistic function distribution learning rather than deterministic optimization.

**Geometric Bases:** They introduce structured priors that act as geometric bases - essentially learned 3D Gaussians with semantic embeddings extracted via transformer attention. These serve as an information bottleneck to encode spatial structure and bridge the gap between 2D context images and 3D target predictions.

**Hierarchical Latent variables:** Instead of a single latent z, they use global (object-level) and local (coordinate/ray-level) variables to balance scene-wide consistency with coordinate-specific predictions.

**Audience:**

Yes

**Claims And Evidence:**

Yes

**Requested Changes:**

See section above

**Strengths And Weaknesses:**

**Strengths**
- The paper tackles an important problem with a well-motivated approach. The geometric bases serve as effective spatial regularizers, helping align sparse 2D observations with 3D target points—a key challenge in few-shot 3D synthesis. The hierarchical latent structure (global + local) prevents information bottlenecks by allowing the model to capture both overall scene structure and coordinate-specific details
- The experimental validation in the paper is comprehensive, demonstrating consistent improvements over baselines across multiple domains (1D signals, 2D images, 3D NeRF synthesis) with notable gains on ShapeNet dataset.
- The ablation studies show the utility of geometric bases and hierarchical latent representations. The qualitative ablations are also interesting, showing how the local and global latent vectors contribute to reconstruction quality.
- The model provides uncertainty quantification which can be useful in applications that require reliability. Their results show that the generated uncertainty maps appropriately highlight edges and boundaries as high-uncertainty regions.

**Weaknesses**
- Although the paper presents the probabilistic generalization of NeF as a core contribution, the formulation in the early part of Section 3 appears quite similar to that of Gu et al.[1]. Articulating the novel aspects of this work more clearly and distinguishing them from prior approaches would significantly improve the presentation.
- The information misalignment problem between 2D context images and 3D discrete points is mentioned a few times but has not been described. Providing a brief explanation could help improve clarity for readers.
- Appendix D.3 has a typo ("We use modulation to The latent variables...").

[1] Gu, Jeffrey, Kuan-Chieh Wang, and Serena Yeung. "Generalizable neural fields as partially observed neural processes." In _Proceedings of the IEEE/CVF International Conference on Computer Vision_, pp. 5330-5339. 2023.

---

> ### Author Response · Authors · 2025-10-16
>
> *We sincerely thank # Reviewer snNU for their insightful comments. The following addresses their concerns and provides answers to their questions. All revisions in the manuscript are highlighted in blue.*
>
> ---
>
> **1. Different formulation from Gu et al.[1].**
>
> Although both our approach and PONP (Gu et al. [1]) follow the neural-process paradigm, PONP encodes all context into a single latent without structural inductive bias, whereas we introduce an explicit hierarchical latent structure tied to geometric bases. By factorizing global and local variables via these bases, we enforce spatial locality and consistency, yielding better reconstruction quality and more calibrated, spatially resolved uncertainty.
> We have clarified this in Section 5.
>
> **2. Clarification on “ information misalignment”.**
>
> By “information misalignment,” we mean that 2D pixels are radiance integrals along camera rays rather than pointwise samples of the 3D field, so occlusion, view-dependence, and depth ambiguity induce a many-to-one, projection-induced mismatch between the observed 2D contexts and the 3D target points. We have clarified this in Section 5.
>
> **3. Typo at Appendix D.3.**
>
> Thank you for pointing out the typo. We have fixed the typo in Appendix D.3.
>
> ---
>
> *Thank you for your time and efforts. We hope the provided clarifications have addressed your concerns. Please don’t hesitate to reach out if you have further questions or need more information.*

---

### Review · Reviewer_H1Qg · 2025-10-08

**Summary Of Contributions:**

This paper introduces and describes a method called Geometric Neural Processes Fields (G-NPF), aimed at improving the generalisation of neural fields. In particular, the contribution of this paper is an extension of neural fields with neural processes to introduce a probabilistic modelling framework, and with geometric bases to encode structural inductive biases. The paper provides background information about the underlying methods, presents the specific contributions of this paper, and offers an empirical analysis of the results with experiments on 1D signal regression, 2D image regression and 3D novel view synthesis.

**Audience:**

Yes

**Broader Impact Concerns:**

This paper does not include a Broader Impact Statement. I would add that not only this section is missing, but also that the paper and the problem it tackles is not sufficiently situated in its broader societal and scientific context, as I discuss in a previous section.

Furthermore, I would highlight that the paper uncritically uses a data set of human faces for training and evaluation. I think it would be important to acknowledge the potential derived impacts and uses of this kind of data.

**Claims And Evidence:**

Yes

**Requested Changes:**

My overall assessment is that the paper has good quality for publication. Nonetheless, there are a few aspects that could be improved, in my opinion, especially in the presentation and discussion of results, as well as in the introduction of the paper. I have detailed my comments and suggestions in the previous section.

**Strengths And Weaknesses:**

As main strengths, I would highlight that this paper is well motivated, well written and my best assessment of it is that it is scientifically sound. I must though mention that while I have read some related works to better understand the context of the paper, I am not very familiar with the literature of neural fields.

In particular, I identify the following aspects as strengths of the paper:

- The background section (Section 2) is well written and it succinctly reviews the related work on neural fields as well as briefly on neural processes, to motivate the introduction of the method presented in this paper.
- The method section (Section 3) is easy to follow. I particularly appreciate that each section builds upon the previous one, offering sufficient context to understand the full proposal.
- The mathematical notation is self-consistent as well as consistent with conventional probabilistic modelling literature.
- Figure 2 (graphical model) is very helpful as a summary of the modelling choices.
- The results seem to support the modelling choices and improve the baselines, although I am not equipped to assess whether the baseline methods are state of the art.
- The results section includes ablation studies about some of the modelling choices.

The weakest part of the manuscript, in my opinion, is the experiments section (Section 4). While careful reading would allow the reader to understand or infer the most important details, I think that the writing could be significantly improved to facilitate the task. For example:

- The introductory paragraph of the section and the sub-sections organise the contents according to data sets and the dimensionality of the data. This makes some sense, but I think it would help to organise the experiments and results in terms of _tasks_ (which are in turn connected to data sets and dimensionalities).
- I would include additional details about the nature of each task, ideally with a mathematical description. For example, while "image regression" may be a canonical task in the neural fields literature, I imagine it is not obvious to everyone what image regression is and why one would care. Similar comments could be made about the other tasks.
- In the image regression task, it is unclear to me how the training and test data are exactly use. I imagine the models are trained with the training data set as usual, but for evaluation, what information is provided to the model to do regression? Is there any re-training at all? How is generalisation to unseen data measured?
- Additional context and details could also be added about other tasks and data sets. For example, the NeRF Synthetic data set is mentioned as though it was common background knowledge. Again, it may be a widely used data set and task within the field, but I think providing some context would enable a wider audience to follow the paper.
- Importantly, I also miss details about the metrics used to evaluate the models. For example, Table 1 includes a set of numbers but no information about what these numbers are. Other tables do include a bit of information about the metrics (an acronym, which should still be unpacked) but in the case of Table 1, I do not think the metrics are described or mentioned anywhere. I think the evaluation metrics should be clearly identified and defined in case they are not obvious.
- While the choice of a probabilistic modelling is theoretically motivated, I would like to see an empirical justification and validation of this approach. I understand that this is a core characteristic of the method which cannot easily be included in the ablation study, but I encourage the authors to find alternative ways to demonstrate the value of this choice.
- Related to the previous point, I think the inclusion of the uncertainty maps and the corresponding section add little value. The discussion is very superficial and the insights are neither sufficiently grounded nor reflect what we see in the maps. In particular, it is not obvious to me why it is expected that the model should have more variance near the edges. As a matter of fact, this not exactly what we see on the images. For example, on the image of the instrument, we see high values far from the edges and in the photo of the church we see low values near sharp edges. On top of this, it is again unclear what the colours are supposed to mean (the label of the colour bar reads in tiny font "Colour Intensity", which is quite uninformative).

Beyond the review of section 4, I would also encourage the authors to improve the abstract and the introduction. This is probably a personal preference of mine that may not be widely shared, but I think papers become more relevant and stronger when they include a discussion of the problem they are trying to solve or the questions they are trying to address in a wider context. For example, the first sentence of the abstract is "This paper focuses on Implicit Neural Representation (INR) generalization, where models need to efficiently adapt to new signals with few observations". First, this assumes that the reader is familiar with INR. Second and personally, I would like to see a couple of sentences about why this matters, why this problem worth studying and how it is connect to the broader societal and scientific context. This information is missing in the abstract, but also the introduction does not provide much more context.

Some additional questions and comments:
- The title of the caption of Table 2 is "Qualitative Comparison". Should it not be "quantitative" instead? Same for Table 3.
- The experimental setup for ShapeNet in Section 4.2 includes sampling random views to evaluate the performance of the models on the task of _novel view synthesis_. I have two questions:
    - Are the views used for evaluation consistent across papers? Are they consistent at least across models evaluated in the paper?
    - Is the evaluation with two randomly (I assume a uniform distribution) sampled views representative of the expected generalisation?
- The formatting of the paragraph titles in Section 4.2 is not consistent (sometimes bold face used, other times italics is used)

---

> ### Author Response · Authors · 2025-10-16
>
> *We sincerely thank # Reviewer H1Qg for their insightful comments. The following addresses their concerns and provides answers to their questions. All revisions in the manuscript are highlighted in blue.*
>
> ---
>
> **1. Introduction paragraph of Section 4.**
>
> Thank you for the suggestion. We have revised the paragraph and organized it by task.
>
> **2. Additional details about the nature of each task.**
>
> Image regression treats an image as a continuous function \\(f:\\mathbb{R}^2 \\to [0,1]^3\\) and learns \\(f\\) from sparse or dense pixel samples to predict color at any queried coordinate, thereby reflecting a model’s capacity to fit and represent a new signal.
>
> 3D novel view synthesis (NVS) learns a continuous radiance field from a limited set of calibrated multi-view images and renders photorealistic images from previously unseen camera poses. This setting requires extrapolating colors along unseen rays while maintaining geometry and appearance consistency, thus providing a stringent benchmark of neural field generalization.
>
>
> 1D signal regression learns a continuous function \\(f:\\mathbb{R} \\to \\mathbb{R}^K\\) from samples \\(\\{(t_i,y_i)\\}\\), where \\(t_i\\in\\mathbb{R}\\) is a 1D coordinate (e.g., time) and \\(y_i\\in\\mathbb{R}^K\\) the target (scalar when \\(K{=}1\\)). Queried at arbitrary \\(t\\) for interpolation or extrapolation, it provides a simple, well-controlled task to quantify extrapolation ability and assess uncertainty calibration.
>
> **3. Image regression clarification.**
>
> We train an amortized encoder on the training set only. For each training image \\(I\\), we sample pixels \\( \\{(x\_{i}, c\_{i})\\}\_{i=1}^{N} \\) with coordinates \\(x_i\\in[0,1]^2\\) and colors \\(c_i\\in[0,1]^3\\). The encoder produces a per-image code $z$ as well as the geometric bases, and the neural field predicts colors via \\(f_\theta(x; z)\\).
>
> At test time, there is **no re-training or adaptation**. For an unseen image from the test set, partial pixels of this image are provided as the context information. Given the observed context pixels, we infer the function (latent variable with geometric bases)  and predict the full image.
>
> Generalization is measured on the unseen test set using peak signal-to-noise ratio (PSNR) over all test images.
>
> We have clarified this in Section 4.
>
> **4. Additional context for the NeRF Synthetic dataset.**
>
> The NeRF-Synthetic dataset contains eight object-centric scenes with complex geometry and realistic materials. For each scene, images are rendered from viewpoints randomly sampled on a hemisphere around the object. We have added this detail in Section 4.
>
> **5. Details about the metrics used.**
>
> We report peak signal-to-noise ratio (PSNR) for both image regression and 3D novel view synthesis tasks. We have clarified this in the revision.
> $$
> \\text{MSE} = \\frac{1}{N}\\sum_{p=1}^{N} \\big(I_p - \\hat{I}_p\\big)^2
> $$
>
> $$\\text{PSNR} = 10 \\log\_{10} \\left(\\frac{\\text{MAX}\_I^{2}}{\\text{MSE}}\\right) = 20 \\log\_{10}(\\text{MAX}\_I) - 10 \\log\_{10}\\text{MSE}$$
>
> **6. Probabilistic evaluation.**
>
> To justify the probabilistic framework, we used Table 5 to ablate the hierarchical latent variables (the core of probabilistic modeling).
> Either object-specific or ray-specific latent variable improves the performance, indicating the effectiveness of the probabilistic modeling. With both latent variables, the method performs best, demonstrating the importance of the hierarchical structure.
>
> **7. Uncertainty map clarification.**
>
> We apologize for any confusion caused by the uncertainty map. Elevated predictive uncertainty typically arises near geometric or reflectance discontinuities, reflecting practical model–likelihood mismatches (e.g., discretization, anti-aliasing, or modest measurement noise). Consequently, residual error concentrates along high-frequency boundaries. In the instrument example, the pronounced off-edge uncertainty is attributable to embedded watermarks: although barely perceptible to human observers, these signals introduce fine-scale texture ambiguities that our model captures as increased predictive variance.
>
>
> Colormap denotes per-pixel predictive variance (unitless, RGB in [0,1][0,1][0,1]); brighter indicates larger variance.
>
> To avoid confusion, we move this figure to Appendix F.7 and additionally provide uncertainty maps for the image-completion setting, where NPs exhibit higher uncertainty under limited context.  As shown in Fig.~16, the uncertainty concentrates in regions with fewer or no nearby context pixels, and it shrinks as the number/coverage of context points increases. With limited context, the model admits multiple plausible completions, resulting in higher predictive spread; with richer context, the model concentrates, yielding sharper, lower-variance maps that correlate with reconstruction error.
>
> We have revised the context and the figures, including the font size, accordingly in  Appendix F.7.

---

> > ### Author Response · Authors · 2025-10-16
> >
> > **8. Abstract and the introduction suggestion.**
> >
> > Thank you for your suggestion. We have revised the abstract and the introduction accordingly.
> >
> > **9. Caption of Table 2.**
> >
> > Thank you for pointing out the typo. We have corrected this typo.
> >
> > **10. Experimental setup for ShapeNet.**
> >
> > (a). Yes, the evaluation views are consistent across papers. In this benchmark, we randomly sample one context view and repeat the entire test set 100 times. Because we use the same random seed and the same procedure at test time, the setting is identical.
> >
> > (b). Yes. This protocol is representative, as the evaluation is repeated 100 times.
> >
> > **11. Paragraph title format.**
> >
> > Thank you for your suggestions. We have revised the paragraph titles in Section 4.2 and made them consistent.
> >
> > ---
> >
> > *Thank you for your time and efforts. We hope the provided clarifications have addressed your concerns. Please don’t hesitate to reach out if you have further questions or need more information.*

---

> > > ### Comment · Reviewer_H1Qg · 2025-10-26
> > >
> > > Thank you for the comments and responses to my questions. These have clarified most of my questions. I also appreciate the changes introduced in the updated manuscript to incorporate the feedback from the reviews.
> > >
> > > As remaining comments:
> > > - I am still not convinced about the arguments regarding the uncertainty maps. I think the argument that the "uncertainty concentrates in regions with fewer or no nearby context pixel" is not really supported by the examples provided. It's hard to tell for sure, but my impression is that in Figure 16 there is high uncertainty on parts of the image with a good number of pixels (especially on the example on the right). I think this hypothesis should be more systematically verified.
> > > - Also, I would highly encourage to use a different colour map than jet, which is [not perceptually uniform and can be misleading](https://datascience.stanford.edu/news/fixing-non-uniform-colormaps-fixthejet).
> > > - I would still recommend adding more context and motivation in the introduction, as discussed in my original review.

---

> > > > ### Author Response · Authors · 2025-10-27
> > > >
> > > > *We sincerely thank # Reviewer H1Qg for their insightful comments. The following addresses their concerns and provides answers to their questions. All revisions in the manuscript are highlighted in blue.*
> > > >
> > > > ---
> > > >
> > > > **1. Uncertainty maps clarification**
> > > >
> > > > Sorry for the confusion. To better demonstrate the uncertainty map, we (a) plot predictive uncertainty versus distance to the nearest context pixel and (b) add a lower-half masked visualization.
> > > >
> > > > For (a), we use the distance to the nearest context pixel to quantify local sparsity (larger distances indicate sparser context), and present its relationship to predictive uncertainty. Specifically, we use quantile binning of the distance values (e.g., 12 bins) so each bin spans a similar fraction of pixels rather than a fixed numeric range. For each bin, we report the mean uncertainty and its standard error and plot these against the bin centers, providing a stable, interpretable trend of how uncertainty evolves as locations move farther from the available context.
> > > > As shown in Fig.16, uncertainty increases with distance to the nearest context pixel.
> > > > For (b), Fig. 17 shows higher uncertainty concentrating in the masked lower half where no context pixels are provided. Most pixels follow the same conclusion that uncertainty increases with distance to the nearest context pixel.
> > > >
> > > > **2. Color map**
> > > >
> > > > Thank you for your suggestion. We have changed to the viridis colormap, which is perceptually uniform.
> > > >
> > > > **3. More context and motivation in the introduction**
> > > >
> > > > Thank you for your suggestion. We have added more context and motivation in the introduction.
> > > >
> > > > Context:
> > > >
> > > > NeFs (Vyas et al., 2024) are now used broadly in image and signal processing, including shape representation (Genova et al., 2020), novel view synthesis (Mildenhall et al., 2021), material rendering (Kuznetsov, 2021), medical imaging (Wang et al., 2022b), linear inverse problems (Sun et al., 2021), and compression (Dupont et al.,
> > > > 2021a).
> > > >
> > > > Motivation:
> > > >
> > > > It is crucial for real-world deployment, where only sparse or constrained observations (common in robotics, AR/VR, and medical/computational imaging) are provided, requiring data-efficient adaptation with calibrated uncertainty to obviate costly per-scene optimization and to guide acquisition.
> > > >
> > > > ---
> > > >
> > > > *Thank you for your time and efforts. We hope the provided visualization and clarifications have addressed your concerns. Please don’t hesitate to reach out if you have further questions or need more information.*

---

> > > > > ### Comment · Reviewer_H1Qg · 2025-10-27
> > > > >
> > > > > Thanks for the additional updates. The plot of the distance to nearest context pixel vs. uncertainty is a good idea and it makes a more convincing case for your conclusion. I have definitely less doubt now. I also think using viridis colour map is a much better choice.

---

### Decision · Action_Editor_FABP · 2026-03-16

**Recommendation:** Accept as is

**Additional Comments:**

Apologies for the delays in the handling of this manuscript.  It is a solid piece of work.

**Audience:**

Yes

**Audience Explanation:**

Neural processes and radiance fields are both topics of interest to those working on computer vision and 3D related applications.

**Claims And Evidence:**

Yes

**Claims Explanation:**

The reviewers and the AE all agreed that the paper is well written and the claims well substantiated by the experimental evaluation.